# Dynamic Model Editing to Rectify Unreliable Behavior in Neural Networks

## Abstract

The performance of neural network models deteriorates due to their unreliable behavior on corrupted input samples and spurious data features. Owing to their opaque nature, rectifying models to address this problem often necessitates arduous data cleaning and model retraining, resulting in huge computational and manual overhead. This motivates the development of efficient methods for rectifying models. In this work, we propose leveraging rank-one model editing to correct model's unreliable behavior on corrupt or spurious inputs and align it with that on clean samples. We introduce an attribution-based method for locating the primary layer responsible for the model's misbehavior and integrate this layer localization technique into a dynamic model editing approach, enabling dynamic adjustment of the model behavior during the editing process. Through extensive experiments, the proposed method is demonstrated to be effective in correcting model's misbehavior observed for neural Trojans and spurious correlations. Our approach demonstrates remarkable performance by achieving its editing objective with as few as a single cleansed sample, which makes it appealing for practice.

## 1 Introduction

Neural network models exhibit unreliable behaviors in adapting to inherent or deliberately introduced data distribution shifts (Arjovsky et al., 2019; Lapuschkin et al., 2019; Gu et al., 2019). Such shifts; resulting from, e.g., spuriously correlated features or backdoor triggers, can misguide a model and alter its behavior from the correct decision-making pathway (Ye et al., 2024; Gu et al., 2019). This compromises model reliability and robustness. Due to the inherent opacity of deep models, primary strategies for correcting such unreliable behavior involve data cleaning and model retraining (Ross et al., 2017; Schramowski et al., 2020; Anders et al., 2022). However, these techniques necessitate both labor-intensive manual data scrutiny and substantial computational overheads (Brown et al., 2020; Achiam et al., 2023; Touvron et al., 2023). Consequently, efficient techniques for correcting unreliable model behaviors emerge as a critical requirement for enhancing their reliability and sustaining the performance of developed models.

This paper investigates efficient correction of unreliable model behavior through rank-one editing (Bau et al., 2020). Originally proposed for editing generative rules encoded by generative models (Bau et al., 2020; Tewel et al., 2023), rank-one model editing has garnered attention for its ability to revise model prediction rules. Expanding on this notion, recent works have adapted this editing approach for domain adaptation in discriminative models (Santurkar et al., 2021; Raunak & Menezes, 2022). However, we formally pinpoint two key challenges when applying rank-one editing to domain adaptation, which inevitably lead to diminished model performance and necessitate labor-intensive data preparation (details in § 4.1). In contrast, we establish that rank-one model editing is well-suited for correcting unreliable model behavior as it intrinsically sidesteps these challenges. To this end, we propose model editing for misbehavior correction with cleansed samples.

Current research on model editing often focuses on editing the deepest feature extraction layer, leveraging its high-level feature encoding capabilities (Santurkar et al., 2021; Raunak & Menezes, 2022). However, our investigation reveals that editing different layers of a model leads to significantly distinct performances. Hence, to locate the layer primarily responsible for the unreliable behavior of the model, we analyze the model's prediction attributions across all its internal layers, comparing predictions for the corrupt samples to those for the cleansed samples. We find that the layer mainly

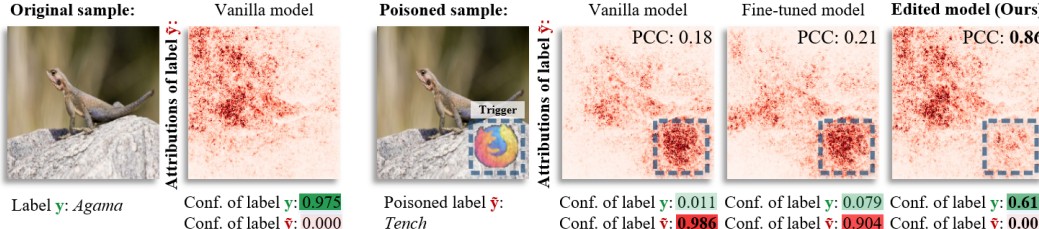

Figure 1: Given the original sample labeled as *Agama*, i.e., class **y**, the Trojaned model can correctly classify this sample. However, it misclassifies the poisoned sample containing a trigger as *Tench*, i.e., class **ȳ**. Attribution maps with Pearson Correlation Coefficients (PCCs) and predictive confidence for the vanilla model, fine-tuned model, and model edited with our approach are provided. Our method restores the correct label by assigning appropriate attributions to the correct object.

responsible for the unreliable behavior can be identified by assessing attributions focusing on the editable parameters of the layer. We introduce a dynamic model editing technique which leverages this layer localization mechanism. Our technique facilitates dynamic selection of the layers during the editing process, further enhancing the efficacy of model editing. Figure 1 shows a representative example showcasing the abilities of our approach in correcting the model decision for a manipulated sample, as evidenced by the attribution maps, Pearson Correlation Coefficients (PCCs), and confidence scores.

The efficacy of our approach is established through experimentation for two well-known model vulnerabilities; namely neural Backdoors/Trojans (Chen et al., 2019b) and spurious correlations (Ye et al., 2024), using CIFAR (Krizhevsky et al., 2009) and ImageNet (Russakovsky et al., 2015) datasets. Experimental evaluations highlight our method's performance, offering an excellent trade-off between model performance and the number of utilized cleansed samples. Notably, our method also achieves high performance with only one cleansed sample. We also extend our assessment to the real-world problem of skin lesion analysis using the ISIC dataset (Codella et al., 2019), thereby illustrating the broader applicability of our approach in practical settings. The key contributions of this paper can be summarized as follows.

1. It introduces the unique concept of leveraging rank-one editing for rectifying model misbehavior resulting from neural Trojans and spurious correlations.
2. It proposes an algorithmic method for suspect layer localization, leveraging the notion of attributions, to identify the primary layer responsible for model unreliabilities.
3. It devises a dynamic model editing framework incorporating the proposed suspect layer localization method. Efficacy of the approach is verified extensively across diverse datasets.

## 2 RELATED WORK

**Unreliable Model Behaviors.** Despite their impressive performance, neural network models have been found to exhibit numerous unreliable behaviors that lead to incorrect predictions on corrupted samples. For instance, the existence of spurious correlations, also known as Clever Hans behavior (Pfungst, 1911), pose a substantial threat to the reliability of these models. A range of spuriously correlated features have been identified including object backgrounds (Xiao et al., 2020), hair color (Sagawa et al., 2019) and colored patches (Gutman et al., 2016). In addition to inherent bias, training data can be intentionally poisoned by mislabeling samples and adding trigger patterns to mislead model predictions (Gu et al., 2019). More attacks (Chen et al., 2019b; Li et al., 2021b; Turner et al., 2019) are proposed to implant invisible triggers for concealed backdoors. Adversarial attacks have demonstrated a significant capacity to alter model predictions (Goodfellow et al., 2015; Madry et al., 2018). However, their practical applicability is often constrained by the necessity of full access to the target model. Consequently, this paper focuses specifically on investigating backdoor attacks and spurious correlations, recognizing their significant impact on undermining the security of deep learning models.

**Model Explaining and Diagnosis.** Various techniques have been proposed to explain and diagnose the vulnerable behaviors of deep models. Attribution methods, such as InputGrad (Simonyan et al.,

2014), GradCAM (Selvaraju et al., 2017) and IG (Sundararajan et al., 2017), assign importance to each input feature to provide explanations for model predictions, which are widely utilized for visually inspecting model behavior (Lapuschkin et al., 2019; Li et al., 2021b). Other efforts (Lapuschkin et al., 2019; Anders et al., 2022) are also made to diagnose unreliable behavior in trained models. For instance, SpRAy (Lapuschkin et al., 2019) analyses heatmaps of training samples to identify Clever Hans behaviors. Anders et al. (2022) proposed A-ClArC and P-ClArC to prevent the propagation of artifact signals. Similarly, the statistics of internal activations are also widely used in revealing backdoor Trojaning (Tran et al., 2018; Hayase et al., 2021; Qi et al., 2022). Despite the availability of various techniques for detecting model unreliability, efficiently and effectively addressing the identified issues remains a significant challenge.

## 3 PRELIMINARY

Model editing (Bau et al., 2020; Meng et al., 2022) focuses on editing a specific model prediction rule while preserving the learned rules. When examining the $l$-th layer of a model $f$, an input sample $x$ is mapped to a feature map $f_l(x)$. The mapped features $f_l(x)$ are recognized for their capability to encapsulate semantic concepts in the representation space (Anderson, 1972; Kohonen, 2012). This understanding is extended to characterize a layer as a linear associative memory. Specifically, assuming a location of the input feature $f_{l-1}(x)$ to be a "key" $k \in \mathbb{R}^n$, the weights $W \in \mathbb{R}^{m \times n}$ within the $l$-th layer map this key $k$ to a "value" $v \in \mathbb{R}^m$ of output features, achieved through the operation $v = Wk$. Considering a finite set of key-value pairs $K = [k_1, k_2, \dots]$ and $V = [v_1, v_2, \dots]$, we can uniquely retrieve a value from a key if the keys are mutually orthogonal. Beyond the exact equality, weight $W$ can be extended to arbitrary non-orthogonal keys by minimizing the error as $W = \arg\min_W \sum_i \|v_i - Wk_i\|^2$. Given this characteristic, Bau et al. (2020) edited model weights to associate a key $k^*$ with a new value $v^*$, effectively rewriting generative model rules.

Recent studies, inspired by the efficacy of the editing technique demonstrated in generative models (Bau et al., 2020; Tewel et al., 2023), apply this paradigm to discriminative models (Santurkar et al., 2021; Raunak & Menezes, 2022). Santurkar et al. (2021) enhanced the domain adaptation capability of classifiers by modifying their prediction rules. For instance, in the case where a "car" classifier struggles to recognize cars featuring "wooden wheels", the model's rules are edited to establish an association between the "wooden wheels" feature and the corresponding activations of "car", enabling the recognition of cars equipped with wooden wheels. While incorporating a new key-value pair, it is critical to ensure the preservation of previously learned associations. Consequently, this editing process is formulated as a constrained least squares problem that creates a new key-value associative memory, and preserves the established key-value associations as

$$\min_\Lambda \|v^* - f_l(k^*; W')\| \quad \text{s.t.} \quad W' = W + \Lambda(C^{-1}k^*)^\top, \tag{1}$$

where $C = KK^\top$ denotes the second moment statistics, and $\Lambda \in \mathbb{R}^m$ is the solution. Since $C^{-1}k^*$ and $\Lambda$ are vectors, the update weights $\Lambda(C^{-1}k^*)^\top \in \mathbb{R}^{m \times n}$ is a rank-one matrix. Hence, the editing process defined by Eq. 1 is termed rank-one editing.

## 4 CORRECTING UNRELIABLE BEHAVIOR WITH MODEL EDITING

In this section, we first pinpoint the intrinsic challenges in leveraging rank-one model editing in its known application of domain adaption. Following that, we propose using it to correct unreliable model behaviors such that these challenges are inherently sidestepped by the proposed technique.

### 4.1 CHALLENGES OF MODEL EDITING

To understand the utility and limitations of rank-one model editing, let us revisit Eq. 1, which defines the target function of rank-one model editing. To preserve the established key-value associations, Eq. 1 updates the model weight $W$ within the space mapped by the matrix $C = KK^\top$, derived from the second-order characteristics of the learned keys $K$. The mapping by matrix $C$ facilitates the decorrelation of a key $k^*$ from the existing keys $k_i \in K$, thereby mitigating interference with the established associative memories during optimization. However, critical challenges arise when the new key $k^*$ is not included in the statistical matrix $C$ for applying rank-one model editing to domain adaption. Specifically, we establish the following lemma.

**Lemma 1.** *For $K = [k_1, k_2, ..., k_d] \in \mathbb{R}^{n \times d}$ and $C = KK^\intercal$, when $k^* \notin K$, the projection $C^{-1}k^*$ leads to a residual component $C^{-1}r$ outside the span of $K$, measurable by a residual vector $r \in \mathbb{R}^n$.*

The proof of Lemma 1 is provided in App. A.1. This lemma highlights that exclusion of the new, unseen key $k^*$ from the set $K$ may adversely affect the preservation of the established key-value associations as $k^*$ does not fall within the span of $K$. This can degrade the overall performance of the edited model. Giving it due importance, we mention this phenomenon as a challenge below.

**Challenge 1.** *Diminished Performance: The exclusion of a key $k^*$ from the statistic matrix $C$ compromises the model's ability to preserve established key-value associations. This omission poses a risk to the performance of the model relying on the established associative memories $K$ and $V$.*

To comprehend rank-one editing limits for domain adaption, we must also consider the disparity in the data distributions involved in the task. An implication of this disparity is that a new key $k^*$ mapping from an unseen sample $x^*$ within a set $\mathcal{X}$, puts an extra burden on data requirements.

**Lemma 2.** *Let $x^* \to k^*$ s.t. $x^* \in \mathcal{X} \sim D'$ and $D' \neq D$, where $D$ is the original data distribution. Then $||k^* - f(x^*; W_D)|| \to 0$ only when $|\mathcal{X}| \gg 0$, where $|.|$ denotes cardinality of the set.*

The proof of Lemma 2 is provided in App. A.1. This lemma emphasizes the necessity of sufficient exposure to the samples from the new distribution $D'$ to accurately approximate $k^*$. Insufficient number of samples can lead to inaccurate representations, which degrades model performance. We note this fact as the following challenge to concisely present our findings.

**Challenge 2.** *Labor-intensive Data Preparation: For an accurate mapping of a new key $k^*$ derived from an unseen sample $x^*$ in domain adaption, an extensive set of annotated samples is required for effective rank-one model editing.*

In summary, the challenges of rank-one model editing in domain adaptation arise from its inability to preserve established associations when new keys fall outside the statistical representation of learned keys, coupled with the need for extensive data to accurately represent keys from unseen distributions.

### 4.2 MODEL EDITING FOR CORRECTING UNRELIABLE BEHAVIOR

We propose leveraging rank-one model editing to rectify a model's unreliable behavior. To that end, we consider two suspect behaviors that result from feature spurious correlation (Pfungst, 1911; Arjovsky et al., 2019) and Neural Trojans (Gu et al., 2019; Chen et al., 2019b).

*Feature Spurious Correlation:* Given an input sample $x \in \mathbb{R}^p$ with label $y \in \mathbb{R}^c$, and a classifier $f : \mathbb{R}^p \to \mathbb{R}^c$, feature spurious correlations occur when the classifier $f$ exploits the spurious correlated features inherent in corrupted samples $\tilde{x}$ to make predictions. While the model classifies $\tilde{x}$ to their correct class $y$, its reliance on the spurious feature results in a flawed decision pathway, rendering it incapable of correctly classifying samples without the irrelevant spurious feature.

*Neural Trojaning:* In contrast to the spurious features inherent in training data, neural Trojaning is executed by injecting a portion of clean samples with a backdoor trigger and modifying their true label $y$ to the incorrect target label $\tilde{y}$. These poisoned samples $\tilde{x}$ are then integrated into the training set to create a poisoned set. After being trained on this poisoned set, a Trojaned model $\tilde{f}$ is highly likely to misclassify input samples containing the trigger to the target label $\tilde{y}$.

The problems of spurious correlation and neural Trojaning are instances of a classifier's unreliable behavior which emerge from relying on non-robust features. To correct such behavior, we advocate the application of rank-one model editing to rectify the established mapping rule between non-robust features and their corresponding activations. When presented with a corrupted sample $\tilde{x}$ that leads the model to exhibit an unreliable behavior, its cleansed counterpart $x$ can guide the model toward the correct prediction pathway. We designate the input feature derived from the corrupted input $\tilde{x}$ as the key $k^*$, and align activations of $k^*$ to the corresponding value $v^*$ mapped from the cleansed sample $x$. We edit the model to make the feature $k^*$ to yield correct activations $v^*$, thereby correcting the model's unreliable behavior.

**Sidestepping the Challenges.** Our proposed process of model editing to correct unreliable behaviors involves the susceptible model that integrates both original samples $x$ and their corrupted counterpart $\tilde{x}$ into the training procedure. For a susceptible model, the training process integrates both clean samples and their corresponding corrupted counterparts. This integration ensures: $C = KK^\intercal$,

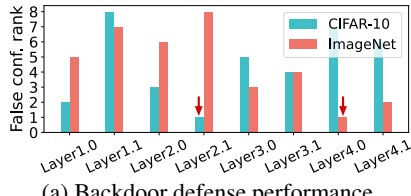

(a) Backdoor defense performance.

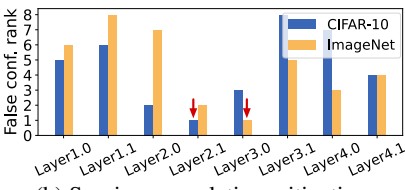

(b) Spurious correlation mitigation.

Figure 2: Performance in reducing false confidence after individually editing different layers of ResNet-18. A lower value indicates better suppression of the model's false confidence. Red arrows indicate the layer yielding the best results for a given dataset after editing.

$K = [k_1, k_2, \ldots, k^*]$, $V = [v_1, v_2, \ldots, v^*]$. This eliminates the residual $r$ such that $C^{-1}k^*$ within the span of $K$. Thus, the unchanged key-value associations preserve model performance, circumventing Challenge 1. By incorporating $\{x, \tilde{x}\} \in \mathcal{X}$ in training, the model ensures $||k^* - f(x^*; W_D)||$ approaching 0 as $x^* \in \mathcal{X}$, when $|\mathcal{X}| \ll 0$ is not available. It mitigates insufficient feature exposure, sidestepping Challenge 2. Thus, repurposing rank-one model editing from domain adaptation to correcting model unreliability effectively sidesteps the inherent challenges, ensuring both *the preservation of model performance* and *the minimization of labor-intensive data preparation*.

## 5 DYNAMIC MODEL EDITING

In this section, we first introduce an attribution-based method aimed at identifying the model layer responsible for its unreliable behavior. The identified layer serves as the foundation for effective model editing. We then integrate this localization technique to construct a dynamic model editing framework, offering an enhanced capability to correct unreliable behavior of a model.

### 5.1 LOCATING SUSPECT LAYER WITH ATTRIBUTION

Rank-one model editing treats convolutional layers as linear associative memories, confining the editing to a specific model layer. Current methods default to utilizing the final convolutional layer for editing (Bau et al., 2020; Santurkar et al., 2021) owing to its capacity to encode high-level input features. However, our investigation reveals a notable variability in the efficacy of model editing when handling different layers. In Fig. 2, we empirically demonstrate that editing performed on distinct model layers can yield significantly diverse results when dealing with unreliable model behavior - experiment details are provided in App. A.5. This motivates the need of a mechanism to locate the suspect layer that is primarily responsible for the observed behavior.

To identify the suspect layer, we leverage integral-based attribution (Sundararajan et al., 2017; Chen et al., 2019a) to quantify the shift in attribution from the model's predictions on corrupted samples $\tilde{x}$ to cleansed samples $x$. Integral-based methods, such as Integrated Gradients (IG) (Sundararajan et al., 2017), calculate the feature attribution by estimating the integral from a designated reference to the input sample. Semantically, the reference signifies absence of the true input feature. This resonates perfectly with the corrupted features in our context. Hence, we define the corrupted input $\tilde{x}$ as the reference to quantify the attributions from $\tilde{x}$ to $x$. We assess the attributions of the change in the final predictions on $\tilde{x}$ and $x$, i.e., $f(x) - f(\tilde{x})$, across all the internal layers in the model $f$. We formulate the attribution $M$ from the prediction on $\tilde{x}$ to $x$ in the $l$-th layer of $f$ as

$$M_i^l(x, \tilde{x}) = (f_l(x_i) - f_l(\tilde{x}_i)) \cdot \int_{\alpha=0}^{1} \frac{\partial f(\hat{x})}{\partial f_l(\hat{x}_i)}\bigg|_{\hat{x}=\tilde{x}+\alpha(x-\tilde{x})} \mathrm{d}\alpha, \quad (2)$$

where $f_l(x_i)$ indicates the $i$-th output feature of the $l$-th layer in $f$, and $\hat{x}$ indicates the interpolated input from the reference input $\tilde{x}$ to the input $x$ along a linear path defined by $\alpha$. Attribution is estimated by accumulating the gradient $\partial f(\hat{x})/\partial f_l(\hat{x}_i)$ of the interpolated inputs.

The attribution maps computed for different internal layers have diverse dimensionalities, which complicates their comparison across the layers. To address this, we leverage the Completeness axiom (Sundararajan et al., 2017) to enable the sought comparability of the attributions. The axiom asserts that the sum of attributions equals the model prediction change from the reference to the

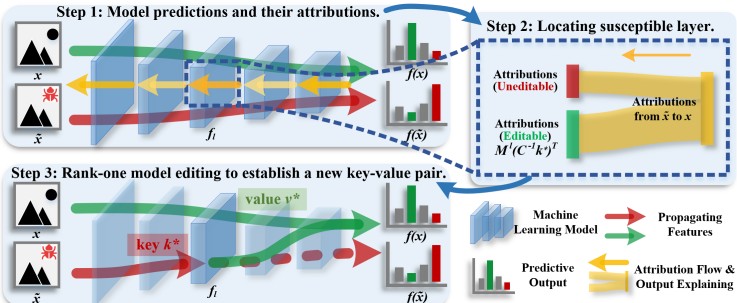

Figure 3: Model editing workflow. Step 1: Given a pair of clean and corrupted samples, their prediction attributions are mapped for all internal layers. Step 2: Attributions are transformed to emphasize editable parameters and locate the suspect layer. Step 3: Rank-one model editing can be applied to establish a new key-value association in the located layer for behavior correction.

input, i.e., $\sum_i M_i = f(\tilde{x}) - f(x)$. We extend this axiom to the internal layers of the model through the following lemma.

**Lemma 3.** *For the $l$-th internal layer $f_l$ of model $f$, $\sum_i M_i^l = f(\tilde{x}) - f(x)$, where $l \in \{1, \ldots, n\}$.*

Proof of Lemma 3 is provided in App. A.1. This lemma establishes that the cumulative attributions of features derived from different layers are consistent. We leverage this fact to systematically treat the attributions of different layers on equal grounds. To elaborate on our computations to identify the suspect layer, let us revisit rank-one model editing defined in Eq. 1. The editing operates in the direction $C^{-1}k^*$ determined by the statistics $C$ of the memorized keys and a new key $k^*$. This implies that the computed attributions need a remapping to identify the editable parameters aligned with the direction $C^{-1}k^*$. We can perform this remapping by the transform $M^* = M(C^{-1}k^*)^\intercal$. Following this, in light of Lemma 3, we employ $||M^*||_F$ to identify the primary suspect layer.

The above computation lays the groundwork for effective model editing. Figure 3 illustrates the pipeline of the proposed layer localization approach in Step 1& 2. Given a corrupted sample $\tilde{x}$ and its cleansed sample $x$, the model yields predictions $f(x)$ and $f(\tilde{x})$ through two distinct decision pathways in Step 1. The prediction change is then attributed to the features derived from different internal layers, quantified by attributions $M^l(x, \tilde{x})$. Calculated attributions are further transformed to emphasize editable parameters by mapping them into the space $C^{-1}k^*$. In Step 2, the editable information of attributions across layers is compared to identify the suspect layer primarily responsible for the model's unreliable behavior.

## 5.2 MODEL EDITING

It is possible to already establish an effective model editing technique by modifying the suspect layer identified in the previous section. We illustrate this in Step 3 in Fig. 3, where by directly applying rank-one model editing to the suspect layer $f_l$, we remap the key $k^*$ from the corrupted sample $\tilde{x}$ to the value $v^*$ derived from the cleansed sample $x$. Though effective, this would be a form of *static* editing, which does not account for the potential model shift during the editing process itself. Recognizing the problem, we propose dynamic model editing that incorporates our layer localization technique to dynamically identify the suspect layers, and improve them. Our technique facilitates automatic adaptation of the model layers for behavior correction.

Algorithm 1 presents the proposed dynamic model editing framework. Given a model $f$, the objective is to correct the model's behavior on a

---

**Algorithm 1:** Dynamic Model Editing

**input** : model $f$, overall budget $\epsilon$, targeted gap $\delta$, corrupted sample $\tilde{x}$, cleansed sample $x$, rank-one model editing $\Omega$, evaluation metric $\zeta$

1 **initialize**: $\epsilon^* \leftarrow 0$, $\delta^* \leftarrow f(x) - f(\tilde{x})$.
2 **while** $\delta^* > \delta$ *and* $\epsilon^* \leq \epsilon$ **do**
       // locate a layer *cf.* § 5.1
3     $l \leftarrow \arg\max_l ||M^{l*}||_F$
       // model editing *cf.* § 4.2
4     $f^* \leftarrow \Omega(f_l, n, \tilde{x}, x)$
5     $\epsilon^* \leftarrow \mathop{\mathbb{E}}_{(x,y)\sim D} \zeta(f(x), y) - \zeta(f^*(x), y)$
6     $\delta^* \leftarrow f^*(x) - f^*(\tilde{x})$
7     **if** $\epsilon^* \leq \epsilon$ **then**
8         $f \leftarrow f^*$
9 **return** $f$

---

corrupted sample $\tilde{x}$ by aligning it with the decision pathway of the cleansed sample $x$. Assuming prediction gap $\delta^* = f(x) - f(\tilde{x})$, we aim to minimize $\delta^*$ to achieve the target gap $\delta$ within an overall budget of $\epsilon$. Specifically, while the current prediction gap $\delta^*$ exceeds the targeted gap $\delta$, and the overall performance degradation $\epsilon$ remains within the tolerated threshold $\epsilon^*$ (Line 2), the algorithm identifies the $l$-th layer responsible for the unreliability on $\tilde{x}$ by comparing the editable components of attributions $M^*$ (Line 3). Subsequently, rank-one editing is applied to establish a new key-value association in the identified layer (Line 4). Following a predefined number of editing epochs $n$, we update the current budget $\epsilon^*$ and gap $\delta^*$ based on the evaluation results of the edited model $f^*$ (Lines 5-6). If the overall performance degradation in $f^*$ remains within the permissible overall budget $\epsilon$, the edited model $f^*$ is preserved, and the editing process continues (Lines 7-8). Otherwise, the edited model $f$ will be returned (Line 9).

**Time Complexity Analysis.** Let $L$ represent the number of layers in the model, $n$ the number of key-value pairs, and $T$ the number of iterations. The complexity is dominated by two components: attribution computation, which involves a forward and backward pass through the model, scaling as $O(A)$; and rank-one editing, which requires calculation around a static matrix $CC^\mathsf{T} \in \mathbb{R}^{n \times n}$ with a cost of $O(n^3)$. The overall time complexity is $O(T \cdot (A + n^3))$. Since the algorithm uses a small number of cleansed samples, $n$ remains small, minimizing the cost of calculating matrix. Moreover, $T$ is typically less than $L$, ensuring the bound $O(T \cdot (A + n^3)) \leq O(L \cdot (A + n^3))$. As a result, the proposed algorithm achieves efficient computational performance.

# 6 EXPERIMENTS

We conduct extensive empirical validation across diverse datasets to assess the efficacy of our proposed methods. Further details of the underlying experimental setups are also available in App. A.3.

## 6.1 EFFICACY AGAINST NEURAL TROJANS

To evaluate the efficacy of our approach, we conduct experiments on Trojaned models using the CIFAR-10 (Krizhevsky et al., 2009) and ImageNet (Russakovsky et al., 2015) datasets. We create a poisoned set by injecting a backdoor trigger into a subset of training samples, simultaneously altering their original labels $y$ to a poisoned target label $\tilde{y}$. Trojaned models $\tilde{f}$ are then established by training on this poisoned set, leading to the misclassification of samples containing the trigger as the target label $\tilde{y}$ - see App. A.3 for further details. We use overall accuracy (OA) and attack success rate (ASR) (Chen et al., 2019b) as the metrics.

Table 1: Editing backdoor vulnerability. Overall accuracy (OA) and attack success rate (ASR) are reported for varying number ($n$) of samples.

| Methods | CIFAR-10 | | ImageNet | |
|---|---|---|---|---|
| | OA ↑ | ASR ↓ | OA ↑ | ASR ↓ |
| Trojaned model | 93.67 | 99.94 | 69.05 | 87.24 |
| Fine-tuned model ($n$=1) | 90.83 | 73.07 | 65.95 | 79.91 |
| Fine-tuned model ($n$=10) | 91.57 | 30.14 | 68.66 | 33.73 |
| Fine-tuned model ($n$=20) | 91.58 | 13.22 | 68.42 | 21.86 |
| Patched model ($n = 20$) | 89.70 | 12.19 | 65.59 | 13.81 |
| P-ClArC ($n$=20) | 89.97 | 6.21 | 65.42 | 8.09 |
| A-ClArC ($n$=20) | 92.53 | 6.32 | 67.17 | 8.73 |
| Stat. edited model ($n$=1) | 92.93 | 2.57 | 67.87 | 3.01 |
| Dyn. edited model ($n$=1) | 93.65 | 1.34 | 66.77 | 1.61 |
| Dyn. edited model ($n$=20) | 93.61 | 0.26 | 68.84 | 0.12 |

**Overall Evaluation.** Table 1 summarizes extensive results on different models, including fine-tuned models, patched models (Wang et al., 2019), models learned by projective and augmentative class artifact compensation methods (P-ClArC and A-ClArC) (Anders et al., 2022). P-ClArC and A-ClArC are originally proposed to suppress and correct model unreliabilities by creating suppressive and inductive artifact modules when applied to corrupted images. Evaluated techniques utilize a specific number of cleansed samples ($n$) collected from the original training set. While P-ClArC significantly reduces the ASR compared to fine-tuned models, it degrades the overall model accuracy. Conversely, A-ClArC, which further retrains the model layers, improves clean accuracy but also results in a slight increase in ASR. Similarly, models patched by pruning backdoor-related neurons experience a decline in overall performance. In contrast, our method significantly reduces ASR with minimal cleansed input samples, while retaining high overall accuracy. In the table, we also include the static variant of our approach, illustrated in Fig. 3, for which we only edit the final layer. It is notable that the models edited dynamically consistently outperform those edited at only the final layer, underscoring the effectiveness of our dynamic editing approach. We also perform visual inspections of attribution maps to correct the model's reliance on backdoor features. This is illustrated in Fig. 1 and further figures in App. A.9.

Table 2: Generalization comparison for trigger in different visibility of 0.3, 0.7 and 1.0. One corrupted sample with trigger visibility of **0.5** used for model patching and editing. ASR at visibility $\varphi$ is denoted as $\Gamma_\varphi$.

| Methods | OA ↑ | $\mathbf{\Gamma_{0.5}}$ ↓ | $\Gamma_{0.3}$ ↓ | $\Gamma_{0.7}$ ↓ | $\Gamma_{1.0}$ ↓ |
|---|---|---|---|---|---|
| Benign model | 92.85 | 95.29 | 95.15 | 97.81 | 99.21 |
| Patched model | 89.61 | 26.86 | 30.84 | 32.42 | 37.19 |
| Dyn. edited model | 91.21 | 5.17 | 6.84 | 7.65 | 7.91 |

Table 3: Generalization comparison for trigger located at top-left (TL), Center (C) and bottom-left (BL). One corrupted sample with trigger at bottom-right (**BR**) used for model patching and editing. ASR at location $\eta$ is denoted as $\Gamma_\eta$.

| Methods | OA ↑ | $\mathbf{\Gamma_{BR}}$ ↓ | $\Gamma_{TL}$ ↓ | $\Gamma_{C}$ ↓ | $\Gamma_{BL}$ ↓ |
|---|---|---|---|---|---|
| Benign model | 91.23 | 99.74 | 99.57 | 99.76 | 99.90 |
| Patched model | 89.22 | 29.31 | 34.42 | 34.58 | 34.88 |
| Dyn. edited model | 90.85 | 6.36 | 9.24 | 9.47 | 8.95 |

**Trade-off Evaluation.** In Fig. 4, we demonstrate the mitigation of false predictive confidence of class $\tilde{y}$ by examining how it changes with variations in the number of utilized cleansed samples ($n$) and the overall accuracy degradation during optimization. Remarkably, our methods exhibit outstanding performance even with a single cleansed sample, while resulting in only marginal overall accuracy degradation. In comparison to the fine-tuned (FT) models, our methods showcase an exceptional balance between mitigating false confidence, preserving overall accuracy, and the requirement of cleansed samples.

**Generalization Evaluation.** We further evaluate the generalization of our approach for addressing neural Trojans involving triggers with varying visibilities and spatial locations. First, we train a Trojaned model using poisoned samples with the trigger at visibility levels of 0.3, 0.5, 0.7, and 1.0. To evaluate how well our method generalizes across different trigger visibilities, we patch and edit the model using a single corrupted sample with a 0.5 visibility trigger. As evidenced in Tab. 2, our method effectively mitigates triggers of various visibility levels when using the fixed visibility trigger, demonstrating superior performance compared to the patched model. Next, we evaluate our method's effectiveness in handling triggers placed at different spatial locations. We train a Trojaned model with triggers located at top-left, top-right, center, bottom-left, and bottom-right positions. We then patch and edit the model with a sample containing a trigger positioned at the bottom-right. Table 3 demonstrates that our method successfully handles neural Trojans with triggers located at different positions, based on input with a fixed trigger location.

## 6.2 EFFICACY IN MITIGATING SPURIOUS CORRELATION

We induce spurious correlations in model $f$ by utilizing class-irrelevant patterns as spurious features. Specifically, we pollute a proportion of samples of class $y$ by attaching patterns to create spurious samples $\tilde{x}$. After training on the dataset including these samples, the model tends to rely on spurious features to predict the correct label for class $y$ samples. In our evaluation, we assess the model performance on two distinct sets of class $y$; namely, the clean set and the spurious set. The latter encompasses samples containing spurious features. Reliable models are expected to yield consistent accuracy across both the spurious and clean sets, as well as on the overall testing set.

Table 4 summarises the results for addressing the spurious correlation problem. The table shows that the benign model heavily relies on spurious features for predictions, resulting in higher accuracy

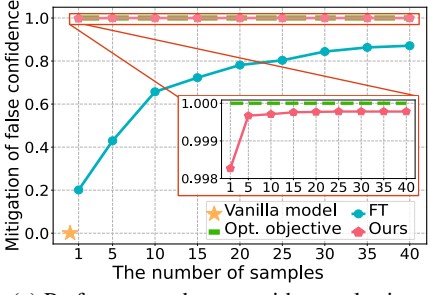
(a) Performance changes with sample size.

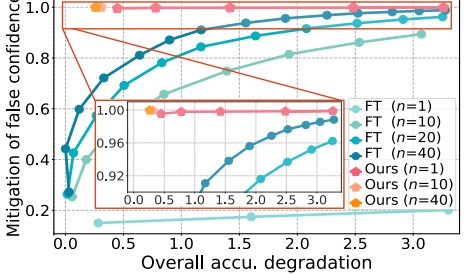
(b) Performance changes with overall accuracy.

Figure 4: Comparison of model performance between fine-tuned models (FT) and edited models by our method (Ours). (a) The mitigation of false confidence changes with the number of used samples, including the vanilla model and the optimization objective. (b) The mitigation of false confidence changes with the overall accuracy degradation (%) during model editing and fine-tuning. Results are computed for ResNet-18 on CIFAR-10 dataset.

on spurious set as compared to the clean set. Fine-tuned models exhibit marginal improvements in mitigating spurious correlations, but may cause even larger absolute performance difference between the clean and spurious sets for the models. A-ClArC, with its inductive module, mitigates spurious correlations but degrades model performance for both sets. Similarly, while P-ClArC shows less disparity between the performances on spurious and clean samples, it leads to unacceptable levels of clean and overall accuracy. In contrast, our approaches demonstrate notable effectiveness in mitigating spurious correlations with a limited cleansed set, yielding model accuracy on spurious set that aligns closely with that on clean set. Moreover, dynamic edited models exhibit heightened efficacy in mitigating spurious correlations.

## 6.3 EVALUATION ON SPURIOUS CORRELATION IN SKIN LESION ANALYSIS

To further assess the broader utility of our method, we applied it to a real-world problem involving skin lesion analysis on the ISIC (International Skin Imaging Collaboration) dataset (Codella et al., 2019). Specifically, we conduct a binary classification of the ISIC data to distinguish between *benign* and *malignant* skin lesions, adhering to the setting of Rieger et al. (2020). In this case, unreliability in the model arises from the presence of colored patches within the benign samples, which introduce spurious correlations learned by the models. Figure 5 shows representative samples from the ISIC dataset, illustrating instances of polluted samples with spurious colored patches. In contrast to the readily available cleansed samples in benchmark datasets like CIFAR and Im-ageNet, acquiring cleansed samples for practical applications is consistently challenging. Thus, we employ a manual approach to remove spurious features by replacing the areas affected by colored patches on the skin with cleaned skin from another region.

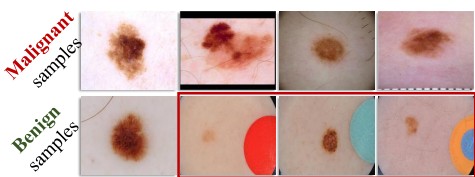

Samples polluted with spurious patches

Figure 5: ISIC samples and the inherent spurious patches. Samples containing malignant and benign lesions from ISIC are presented, where benign samples are partly polluted with spurious colored patches.

In Tab. 5, we present a comparative analysis between fine-tuned (FT) models, A-ClArC and our proposed methods in mitigating the spurious correlation observed in EfficientNet-B4 models (Tan & Le, 2019) trained on the ISIC dataset. Notably, our methods effectively reduce the model's reliance on spurious features with fewer cleansed samples (n=10). Conversely, the fine-tuned model and A-ClArC demonstrate inferior performance and rely on a greater number of cleansed samples. This efficacy in addressing spurious correlations in skin lesion analysis highlights the broad applicability of our method in practical scenarios.

Table 5: Performance comparison for mitigating spurious correlation on ISIC dataset. For our edited models, we use n=10.

| Methods | Overall ↑ | Clean ↑ | Spurious |
|---|---|---|---|
| Benign model | 79.00 | 61.50 | $87.50_{+26.00}$ |
| FT model (n=10) | 79.50 | 62.00 | $83.00_{+21.00}$ |
| FT model (n=20) | 80.50 | 53.00 | $64.50_{+11.50}$ |
| A-ClArC (n=20) | 79.50 | 54.50 | $59.50_{+5.00}$ |
| Stat. edited model | 79.50 | 60.00 | $64.50_{+4.50}$ |
| Dyn. edited model | 80.00 | 61.00 | $62.50_{+1.50}$ |

Table 4: Performance comparison for mitigating spurious correlation on CIFAR-10 and ImageNet. Accuracy (%) is reported for the overall testing, clean and spurious sets. The erroneously increased accuracy on the spurious set, compared to the accuracy on the clean set for samples without the spurious correlated features, is highlighted in red. Smaller increases in accuracy values indicate more desirable outcomes.

| Methods | CIFAR-10 | | | ImageNet | | |
|---|---|---|---|---|---|---|
| | Overall ↑ | Clean ↑ | Spurious | Overall ↑ | Clean ↑ | Spurious |
| Benign model | 94.00 | 94.42 | $100.00_{+5.58}$ | 69.04 | 81.25 | $91.66_{+10.41}$ |
| Fine-tuned model (n=10) | 93.32 | 88.22 | $99.66_{+11.40}$ | 68.01 | 64.58 | $74.99_{+10.41}$ |
| Fine-tuned model (n=20) | 93.47 | 88.97 | $99.62_{+10.65}$ | 68.18 | 64.58 | $74.99_{+10.41}$ |
| P-ClArC (n=20) | 88.29 | 16.89 | $17.12_{+0.23}$ | 66.84 | 8.32 | $10.91_{+2.59}$ |
| A-ClArC (n=20) | 92.41 | 76.77 | $79.34_{+2.57}$ | 67.01 | 75.66 | $82.25_{+6.59}$ |
| Stat. edited model (n=1) | 93.19 | 96.65 | $98.88_{+2.23}$ | 67.64 | 81.25 | $87.50_{+6.25}$ |
| Dyn. edited model (n=1) | 92.93 | 94.29 | $96.15_{+1.86}$ | 67.50 | 81.66 | $85.83_{+4.17}$ |
| Dyn. edited model (n=20) | 93.99 | 94.30 | $94.42_{+0.12}$ | 68.94 | 81.25 | $83.33_{+2.08}$ |

Examples of manually cleaned samples used for model fine-tuning and editing can be found in App. A.3.3. Additional experiments and the evaluation regarding the effectiveness of the proposed layer localization technique are also reported in Apps. A.6 & A.8.

## 7 Limitations and Discussion

In this work, we propose an effective method for efficiently correcting a model's unreliable behaviors. Despite its demonstrated efficacy across diverse scenarios, our approach depends on the identification of unreliabilities and necessitates the availability of both corrupted and cleansed samples. In this section, we examine these limitations within the framework of existing robustness techniques and explore how they relate to broader challenges in deep learning models.

*Comparison with Backdoor Defense Methods.* Current research on backdoor defenses focuses on identifying and neutralizing Trojans embedded within deep models (Li et al., 2021a; Tian et al., 2022). The prevailing strategies to mitigate backdoor attacks involve model retraining and pruning (Liu et al., 2017; Huang et al., 2022). However, these methods are often constrained by the high computational cost of recreating a clean model and the degradation in the model's accuracy on clean data. Furthermore, similar challenges are observed in the field of spurious correlations (Lapuschkin et al., 2019; Anders et al., 2022), where existing methods struggle to efficiently correct models' unreliable behaviors. In contrast, our approach utilizes rank-one model editing to mitigate backdoor attacks, addressing inherent challenges with both efficiency and effectiveness.

*Identification of Unreliability.* While detecting anomalous or Trojaned images is typically addressed as a separate task (Qiao et al., 2019; Huang et al., 2020; Ye et al., 2024), our approach offers several practical advantages by addressing the identification of unreliability in two critical aspects. First, it requires only a single pair of corrupted and cleansed samples to effectively correct the model's behavior. This makes it particularly valuable in scenarios where access to large, cleansed datasets is limited, enabling robust model editing even under resource constraints. Second, our method facilitates image-level correction without the need for precise identification of backdoor triggers or spurious features. By bypassing the need for exact identification of these elements, our approach significantly reduces the complexity associated with pixel-level image cleansing. This adaptability is crucial in practical applications where the availability of original, clean samples is restricted. As a result, our approach allows for efficient model patching even with only coarse detection of inconsistencies or anomalies, making it suitable for a broad range of real-world scenarios.

In summary, our method introduces a robust and scalable paradigm for correcting unreliable behaviors in deep learning models, offering broad applicability across various domains while eliminating the need for precise feature identification or extensive cleansed samples. The scope of this paper is currently limited to image-based experiments. Future work can extend our method to other data modalities. To address existing limitations, future focus on developing model diagnosis and data cleansing framework integrates with the proposed editing technique. This integrated approach will enhance the method's applicability, enabling it to autonomously address a wider range of model deficiencies. Additionally, while the ability to repeatedly edit a fixed layer has been explored in previous work Gupta et al. (2024), the proposed dynamic layer localization method extends this concept to the entire model, which also represents a promising direction for further research.

## 8 Conclusion

In this paper, we first establish that rank-one model editing is well-suited for model misbehavior correction, circumventing the challenges inherent in existing application of domain adaption. We advocate applying the model editing technique to correct model unreliabilities by aligning the model's decision pathways of corrupted inputs with those observed on cleansed inputs. We also introduced an effective attribution-based layer localization method, facilitating the identification of the primary suspect layer for the model's observed misbehavior. We then developed a dynamic model editing framework capable of dynamically adjusting the model for behavior correction. Extensive empirical validation demonstrates remarkable performance of our framework across various scenarios. Particularly noteworthy is the fact that our editing technique requires only a single cleansed sample to achieve high performance levels, which portends its wide applicability in practical scenarios.

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

# A  APPENDIX

## A.1  PROOF

In this section, we provide the proof of Lemmata 1-3. We begin with the proof of Lemma 1

*Proof of Lemma 1.* Consider the key set $K = \{k_1, k_2, \ldots, k_n\} \in \mathbb{R}^{d \times n}$ and the corresponding statistics matrix $C = KK^\top \in KK^\top$. Given a new key $k^* \in \mathbb{R}^d$, the projection of $k^*$ onto the span of $K$ is given by

$$\hat{k} = C^{-1}k^*. \tag{3}$$

The projection $\hat{k}$ is the solution to the following least squares problem by

$$\arg\min_{\beta} ||k^* - K\beta||_2^2, \;\; \beta \in \mathbb{R}^n \tag{4}$$

The solution to this optimization problem is explicitly given by

$$\hat{k} = K(K^\top K)^{-1}K^\top k^* = C^{-1}k^*. \tag{5}$$

If $k^*$ is not in the span of $K$, the projection $\hat{k}$ does not perfectly align with the original key $k^*$. Assume that this misalignment can be quantified by the residual vector $r$, defined as $r = k^* - \hat{k}$. We can express $C^{-1}k^*$ as

$$C^{-1}k^* = C^{-1}\hat{k} + C^{-1}r, \tag{6}$$

which represents the component of $k^*$ that lies outside the span of $K$.

Thus, the exclusion of $k^*$ from the statistic matrix $C$ introduces a residual misalignment in the optimization direction. This misalignment, represented by $r$, interferes with the preservation of existing associative memories, undermining the performance of edited model. $\square$

Below, we provide the proof of Lemma 2.

*Proof of Lemma 2.* Consider a model trained on distribution $D$ with parameters $W$, the key $k^*$ is derived from a new sample $x^* \sim D'$, where $D'$ is a shifted distribution relative to $D$. The model's representation of $k^*$ can be expressed as

$$z^* = f(x^*; W). \tag{7}$$

Since $W$ is optimized for $D$, the representation $f(x^*; W)$ will exhibit bias due to the shift from $D$ to $D'$.

The representation error can be quantified as

$$\epsilon = ||k^* - f(x^*; W_D)||, \tag{8}$$

where $W_D$ are the model parameters trained on $D$. The error $\epsilon$ reflects the divergence between the distributions $D$ and $D'$, given by the KL divergence $\text{KL}(D'||D)$. If the model has not been exposed to sufficient samples from $D'$, this error remains significant.

To rescue $\epsilon$, additional samples $x'^m_{i\,i=1} \sim D'$ are needed. The number of samples $m$ required to accurately learn $k^*$ can be bounded as

$$\mathcal{O}\left(\frac{\text{Var}(x^*)}{\epsilon^2}\right), \tag{9}$$

where $\text{Var}(x^*)$ is the variance of the samples drawn from $D'$. Without sufficient $m$, the model's updated key-value memory will fail to capture the true characteristics of $k^*$, resulting in an inaccurate representation.

Thus, as the number of samples from $D'$ increases, the accuracy of the model's representation of $k^*$ improves. $\square$

*Proof of Lemma 3.* Consider the $l$-th layer $f_l$ of model $f$. The attribution of the $i$-th output feature map derived from $l$-th layer $f_l(x)$ for output prediction change $f_l(x) - f_l(\tilde{x})$ is calculated as

$$M_i^l(x, \tilde{x}) = (f_l(x_i) - f_l(\tilde{x}_i)) \cdot \int_{\alpha=0}^{1} \left.\frac{\partial f(\hat{x})}{\partial f_l(\hat{x}_i)}\right|_{\hat{x}=\tilde{x}+\alpha(x-\tilde{x})} d\alpha. \tag{10}$$

Here, functions $f$ are continuous on the closed interval defined by $\hat{x} = \tilde{x} + \alpha(x - \tilde{x})$, where $\alpha \in [0, 1]$ serves as a parameter along the internal path. Thus, according to the fundamental theorem of calculus for path integrals, the sum of the calculated attributions $M^l$ is equal to the output change $f(x) - f(\tilde{x})$. Formally, this relation can be expressed as

$$\sum_i M_i^l(x, \tilde{x}) = \sum_i \int_{\tilde{x}}^{x} \frac{\partial f(x)}{\partial f_l(x_i)} dx = f(\tilde{x}) - f(x). \tag{11}$$

Thus, we conclude that $\sum_i M_i^l = f(\tilde{x}) - f(x)$ holds for all layers $l \in \{1, \ldots, n\}$. $\square$

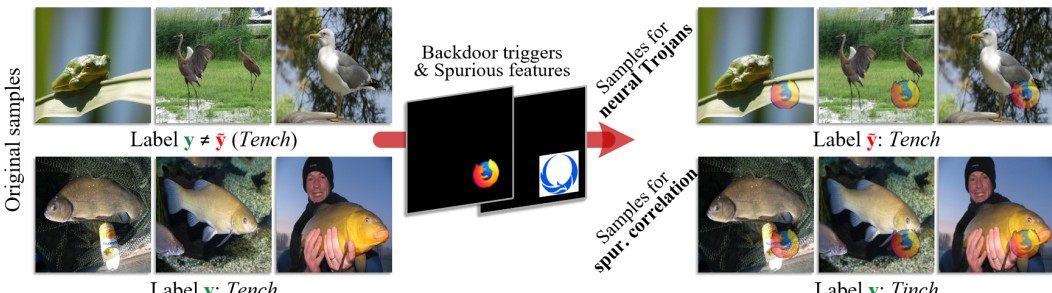

Figure 6: Illustration of samples utilized for neural Trojans and spurious correlations. Two patterns serve as backdoor triggers and spurious features. **Top row**: For neural Trojans, original samples $x$ with label $y \neq \tilde{y}$ are attached with a trigger and changed its label to the target label $\tilde{y}$. **Bottom row**: To induce spurious correlations, samples $x$ of a class $y$ are polluted with spurious features.

### A.2 ZERO-PHASE COMPONENT ANALYSIS IN MODEL EDITING AND LOCATING

In our research, we utilize ZCA (Zero-phase Component Analysis) whitening to enhance the decorrelation of the new key $k^*$ from the established keys $K$, as previously described by Bau et al. (2020). This process involves utilizing a decorrelation matrix $Z = C^{-1/2}$ to further reduce the correlation between the key $k^*$ and the existing keys $K$ by through the transformation $Zk^*$. Let $P$ denote the probability distribution of features at layer $l - 1$, and $K$ represent a discrete distribution over $t$ context examples provided by the user. We measure the information contained in $K$ using cross-entropy $H(K, P)$, akin to the message length in a code optimized for the distribution $P$. In our model, $P$ is assumed to follow a zero-centered Gaussian distribution with a covariance matrix $C$. By normalizing with the ZCA whitening transform $Z$, $P$ can be expressed as a spherical unit normal distribution $P(k) = (2\pi)^{-n/2} e^{-k^\top C^{-1} k/2}$ in the transformed variable $k' = Zk$. This transformation allows us to succinctly express cross-entropy using matrix traces.

Through the normalization of the basis using the ZCA whitening transform $Z$, we transform the probability distribution $P$ into a spherical unit normal distribution, characterized by the variable $k' = Zk$, which enables a compact matrix trace expression for cross-entropy. Leveraging the eigenvector decomposition $C = U\Sigma U^\top$, where $U$ represents the matrix of eigenvectors and $\Sigma$ is the diagonal matrix of eigenvalues, the expression for $Z$ is given by

$$Z = C^{-1/2} = U\Sigma^{-1/2} U^\top. \tag{12}$$

This approach facilitates the decorrelation of the key $k$ through ZCA whitening, effectively implemented as $k = Zk$. In addition, we utilized the computed $Z$ for locating the susceptible layer as described in Section 5.1. Specifically, we map the attributions to focus on editable parameters as $M^* = ZM$.

### A.3 EXPERIMENTAL SETUP

In this section, we provide the comprehensive experimental setup and hyperparameter choices used for model training, model editing and model fine-tuning in our experiments.

#### A.3.1 MODELS

**Trojaned Models.** In this paper, we establish Trojaned models using the blend attack (Chen et al., 2019b). To ensure that the poisoned samples closely resemble the original data distribution, we incorporate the watermark trigger to enhance the backdoor attack. This watermark trigger $\tau$ is defined by $\tau^{(\varphi)} = \varphi \cdot \tau + (1 - \varphi) \cdot x \odot S$, where $\varphi \in [0, 1]$ controls the trigger visibility, and $S \in \{0, 1\}^n$ serves as the mask of trigger $\tau$. In our experiments, the trigger visibility $\varphi$ is set to 0.5. The top row of Fig. 6 illustrates the samples used for model Trojaning. In our experiments, we utilize two trigger patterns to generate poisoned samples. Specifically, evaluations of the models

trained with the Firefox logo are reported in the main paper. Additional experiments involving models trained with the Phoenix logo are detailed in App. A.6.

For Trojaned models trained on ImageNet (Russakovsky et al., 2015), we trained ResNet-18 models with an initial learning rate of $0.1$ for a total of 90 epochs, with the learning rate reduced by a factor of $0.1$ at the 30-th epoch and 60-the epoch. For Traojned models trained on CIFAR-10 (Krizhevsky et al., 2009), we trained ResNet-18 models with an initial learning rate of $0.1$ for a total of 100 epochs, with the learning rate reduced by a factor of $0.1$ at the 50-th epoch and 75-th epochs. For all the Trojaned models under comparison, we choose the first class as the target label $y^*$ for single target Trojaning followed by Qi et al. (2022). On ImageNet, we poison $0.1\%$ of training samples $x$ with label $y \neq y^*$ to embed the backdoor trigger. For CIFAR-10, we set the poisoning rate of $1\%$.

**Models with Spurious Correlation.** To establish models with spurious correlations, we employ trigger patterns as spurious correlated features. The bottom row of Fig. 6 illustrates training samples utilized for inducing model spurious correlation. The training settings for these models are consistent with those used for the Trojaned models. On both ImageNet and CIFAR-10 datasets, we select the first class of samples to induce spurious correlations. For models trained on ImageNet, we contaminate $60\%$ samples of the first class to induce spurious correlation. For models trained on CIFAR-10, we set the contamination rate at $50\%$ for the first class to induce model spurious correlation.

**Models on ISIC.** For models trained on the ISIC dataset, we utilized EfficientNet-B4 models (Tan & Le, 2019). The training process involved using a batch size of $24$ and an initial learning rate of $1 \times 10^{-5}$. The training was conducted over a total of 90 epochs, with the learning rate decaying by a factor of $0.1$ at the 60-th epoch.

### A.3.2 RATIONALE FOR SELECTING THE BLEND ATTACK

In this work, we adopt the blend attack Chen et al. (2019b) to train Trojaned models and spurious correlation-based models. The blend attack was selected for evaluation due to its well-established effectiveness as a backdoor attack strategy. Unlike more recent attack methods Turner et al. (2019); Tian et al. (2022); Nguyen & Tran (2021) that prioritize stealth through minimal perturbations, the blend attack directly integrates triggers into the input, ensuring a substantial impact on the model's predictions. This property makes the blend attack a particularly severe threat, as it strongly biases the model's output toward a predefined target class. By demonstrating robustness against such a potent attack, our method provides compelling evidence of its efficacy. Furthermore, the blend attack's balance between potency and detectability suggests that our approach would generalize effectively to newer or more sophisticated attacks that trade off between these factors.

### A.3.3 MODEL EDITING

**ImageNet and CIFAR-10.** For the ImageNet and CIFAR-10 datasets, we allocate an overall performance budget of 3% accuracy and a tolerated accuracy gap of 0.1% for model editing. For spurious correlations, the overall performance budget is set to 7% accuracy with a tolerated robustness gap of 1% accuracy. The original and corrupted samples used for model editing are depicted in Fig. 6. We utilize an editing learning rate of $1 \times 10^{-4}$ with a weight projection frequency of 10. Unlike other approaches, we do not employ masks to restrict the edited region. Instead, we edit the model at the image level to avoid the need for additional annotations.

**ISIC.** For the ISIC dataset, we set an overall performance budget of 5% accuracy and a tolerated robustness gap of 1% accuracy. The editing learning rate is $1 \times 10^{-5}$ with a weight projection frequency of 10. The editing process is performed at the image level. Unlike datasets that are deliberately created, the ISIC dataset contains corrupted samples from practical scenarios. Consequently, we manually clean these samples by covering the patches with skin tissue from unpolluted regions, as illustrated in Fig. 7.

### A.3.4 MODEL FINE-TUNING

For the model fine-tuning, we retrain only the last convolutional layer of the model while keeping the parameters of the remaining layers fixed. For both ImageNet and CIFAR-10, the learning rate for fine-tuning is set to 0.001. For models trained on the ISIC dataset, the learning rate is set to

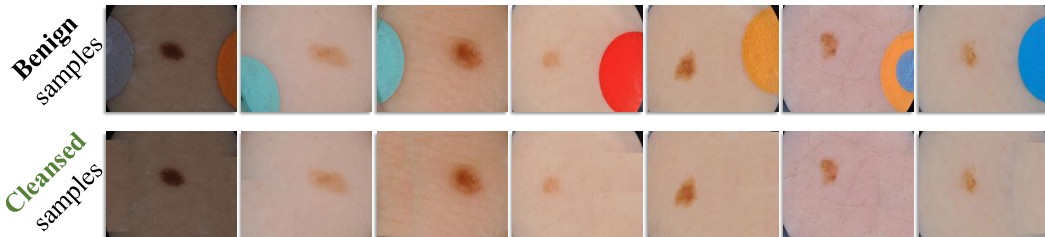

Figure 7: Illustration of cleansed samples on ISIC. For benign samples polluted with colored patches, we manually clean them by covering the patches with skin tissue from unaffected regions.

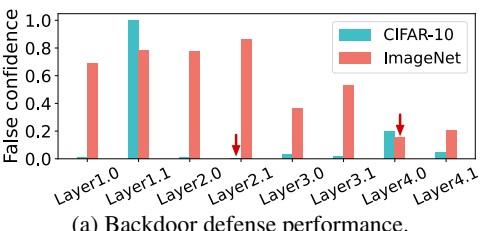
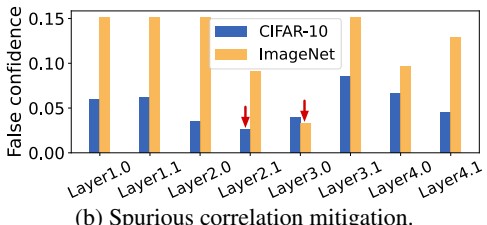

(a) Backdoor defense performance.   (b) Spurious correlation mitigation.

Figure 8: Performance in reducing false confidence after individually editing different layers of ResNet-18. A lower value indicates better suppression of the model's false confidence. Red arrows indicate the layer yielding the best results for a given dataset after model editing.

$1 \times 10^{-5}$. In our experiments, we apply the same budget settings for model fine-tuning as those used for model editing.

### A.3.5 ATTRIBUTION

In this work, we extend the Integrated Gradients method to estimate the attribution difference between cleansed and corrupted samples. Specifically, we approximate the integration defined in Equation 2 in a discrete form as

$$M_i^l(x, \tilde{x}) = (f_l(x_i) - f_l(\tilde{x}_i)) \cdot \sum_{i=1}^{n} \frac{\partial f(\hat{x})}{\partial f_l(\hat{x}_i)}\bigg|_{\hat{x}=\tilde{x}+\frac{i}{n}(x-\tilde{x})} \mathrm{d}\alpha, \qquad (13)$$

where the integration $M_i^l(x, \tilde{x})$ is estimated by integrating the gradients of the interpolated input $\hat{x}$, with $i$ indicating the number of steps. To improve computational efficiency, we leverage recent advancements in Monte Carlo estimation to avoid gradient computations over multiple steps (Erion et al., 2021). Speicifcally, we set $n = 2$, which enhances efficiency while maintaining accuracy.

### A.4 EXPERIMENTAL PLATFORM

All experiments were conducted on a Linux machine equipped with an NVIDIA GTX 3090Ti GPU with 24GB of memory, a 16-core 3.9GHz Intel Core i9-12900K CPU, and 128GB of main memory. The models were developed and tested using the PyTorch deep learning framework (v1.12.1) within the Python programming language. This setup facilitated the efficient handling of computationally intensive tasks, providing a robust environment for both model training and evaluation.

### A.5 EXTENDED EXPERIMENTS OF EDITING DIFFERENT LAYERS

We provide detailed experimental results from applying model editing to different layers of ResNet-18. Using the experimental setup detailed in A.3.3, we independently edited eight distinct layers of ResNet-18 across both CIFAR-10 and ImageNet datasets. For each dataset, eight separate edited models were generated, allowing us to systematically assess the impact of modifying different internal layers. Figure 8 illustrates the results of individually editing different internal layers of ResNet-18 against backdoor attacks and spurious correlations. It is observed that models trained on different

Table 6: Performance comparison of defending against the backdoor attack on Trojaned models trained with the Pheonix logo on CIFAR-10 and ImageNet. Overall accuracy (%) and attack success rate (ASR) are compared between fine-tuned models and models edited by our methods.

| Method | CIFAR-10 | | ImageNet | |
|---|---|---|---|---|
| | Overall Accu. ↑ | ASR ↓ | Overall Accu. ↑ | ASR ↓ |
| Trojaned model | 94.01 | 99.79 | 68.95 | 78.24 |
| Fine-tuned model (n=1) | 91.59 | 69.07 | 65.45 | 77.45 |
| Fine-tuned model (n=20) | 92.85 | 9.70 | 68.63 | 20.23 |
| Edited model (n=1) | 93.32 | 4.49 | 66.06 | 15.24 |
| Dynamic edited model (n=1) | 93.37 | 0.65 | 66.74 | 6.15 |
| Dynamic edited model (n=20) | 93.55 | 0.16 | 68.86 | 1.73 |

Table 7: Performance comparison of mitigating spurious correlation on susceptible models trained with the Pheonix logo on CIFAR-10 and ImageNet. Accuracy (%) is reported for the overall testing set, clean set and spurious set. To facilitate comparison, we present the increased accuracy on the spurious set relative to the accuracy on the clean set.

| Method | CIFAR-10 | | | ImageNet | | |
|---|---|---|---|---|---|---|
| | Overall ↑ | Clean ↑ | Spurious | Overall ↑ | Clean ↑ | Spurious |
| Benign model | 94.14 | 94.67 | 97.15 $_{+2.48}$ | 69.14 | 77.08 | 95.83 $_{+18.75}$ |
| Fine-tuned model (n=10) | 93.67 | 86.80 | 93.93 $_{+7.13}$ | 67.41 | 65.99 | 89.24 $_{+23.25}$ |
| Fine-tuned model (n=20) | 94.07 | 86.67 | 93.28 $_{+6.61}$ | 67.83 | 68.32 | 85.72 $_{+17.40}$ |
| Dyn. edited model (n=1) | 94.03 | 93.28 | 94.78 $_{+1.50}$ | 66.19 | 93.35 | 86.42 $_{+6.93}$ |
| Dyn. edited model (n=20) | 94.04 | 97.15 | 97.89 $_{+0.74}$ | 67.60 | 81.25 | 84.08 $_{+2.83}$ |

tasks and datasets exhibit distinctive effectiveness in reducing false confidence after editing model layers. Moreover, the optimal order of layers for achieving the best mitigation of false confidence differs across these models. This variation underscores the critical need for an effective layer localization technique that can identify which layers should be targeted for editing.

## A.6 Extended Experiments

In this section, additional experimental results are provided for models trained with the Phoenix logo.

**Efficacy in Defending Against Neural Trojans.** Tab. 6 presents a comparison of the performance of Trojaned models, fine-tuned models, and edited models on both CIFAR-10 and ImageNet datasets. The experimental results demonstrate that the proposed model editing technique yields outstanding performance, effectively defending against the backdoor attack. In comparison to fine-tuned models, models edited using our techniques achieve a remarkable trade-off between overall accuracy degradation and the decrease in attack success rate, while requiring only a few cleansed samples.

**Efficacy in Mitigating Spurious Correlations.** In Tab. 7, we assess the effectiveness of our techniques in mitigating spurious correlations on CIFAR-10 and ImageNet. The comparison demonstrates that our method effectively mitigates reliance on spurious features. In contrast to fine-tuned models, which exhibit decreased accuracy on both clean and spurious sets, our techniques enable an increase in accuracy on the clean set. Furthermore, our technique also leads to significant performance improvements with the increased number of cleansed samples, highlighting its superiority.

## A.7 Extended Experiments on Waterbirds dataset

In Table 8, we present a comparative analysis of the performance of a ResNet-34 model trained on the Waterbirds dataset Sagawa et al. (2019). This dataset is known for introducing a bias by relying on spurious background features to distinguish between landbirds and waterbirds. To evaluate the effectiveness of our approach, we compare models trained using Group GRO Sagawa et al.

Table 8: Performance comparison for mitigating spurious correlation on Waterbirds dataset. The accuracy values (%) for both the worst group and the entire dataset are reported.

| Method | Worst-Group Accuracy | Overall Accuracy |
|---|---|---|
| Benign Model | 62.90 | **87.70** |
| Group DRO | 63.60 | 87.60 |
| Fine-tuned model (n=10) | 63.12 | 86.50 |
| Edited model (n=10) | 66.84 | 87.64 |
| Dyn. Edited model (n=10) | **69.18** | 87.68 |

(2019), models fine-tuned to reduce bias, and models edited using our proposed method. The results highlight that our method substantially reduces the model's dependence on these spurious features, leading to a significant improvement in performance. Notably, our approach achieves these gains with a smaller number of cleansed samples (n=10), demonstrating both efficiency and robustness in mitigating the impact of spurious correlations. These findings suggest that our method offers a promising direction for improving the interpretability and generalization of models trained on biased datasets.

## A.8 EVALUATION OF LAYER LOCALIZATION TECHNIQUE

In this section, we evaluate the effectiveness of the proposed layer localization technique. We train 5 ResNet-18 models with 8 internal layers on CIFAR-10, ImageNet, and the ISIC dataset, utilizing two different trigger patterns. Similarly, we establish 5 ResNet-34 models with 16 internal convolutional layers on these three datasets. Additionally, we train 2 EfficientNet-B4 models on both CIFAR-10 and the ISIC datasets, focusing on the 12 internal layers with a kernel size of 3. For the evaluation, we separately edit different internal layers and assess the performance of the edited models. We rank their performances to establish the ground truth for evaluating the recall rate of the located layers. Table 9 presents the recall rates for the top-1, top-3, and top-5 located layers. The results demonstrate that our localization technique achieves high recall rates, effectively identifying the susceptible layers.

Table 9: Results of recall rate (%) in using the proposed susceptible layer localization technique on ResNet-18, ResNet-34 and EfficientNet-B4 models.

| Method | Top-1 Recall ↑ | Top-3 Recall ↑ | Top-5 Recall ↑ |
|---|---|---|---|
| ResNet-18 | 80% | 100% | 100% |
| ResNet-34 | 80% | 80% | 100% |
| EfficientNet-B4 | 50% | 100% | 100% |

## A.9 VISUAL INSPECTION BY ATTRIBUTIONS

**Visual Inspection in Defending Against Backdoor Attacks.** In Fig. 9, we provide additional visual inspection by attribution methods (Sundararajan et al., 2017). Given the original sample $x$ with label $y \neq y^*$, the vanilla model misclassifies the poisoned samples $\tilde{x}$ into the target class $y^*$. Compared to the fine-tuned model, the proposed dynamic model editing technique can effectively correct this unreliable behavior in the deep model, restoring the attribution maps to align with those derived from the original samples.

**Visual Inspection in Mitigating Spurious Correlations.** Figure 10 presents the comparison of attribution maps derived from the vanilla model, fine-tuned model, and models edited using our method. We can observe that our approach effectively mitigates the false reliance on spurious correlated features of the Firefox logo, aligning the attribution maps with those of the original samples.

Figure 11 illustrates the attribution maps for the vanilla model, fine-tuned model, and dynamically edited model. It can be observed that our method effectively corrects the model's reliance on spuriously correlated features in corrupted samples, aligning the attribution maps with those of the cleansed samples.

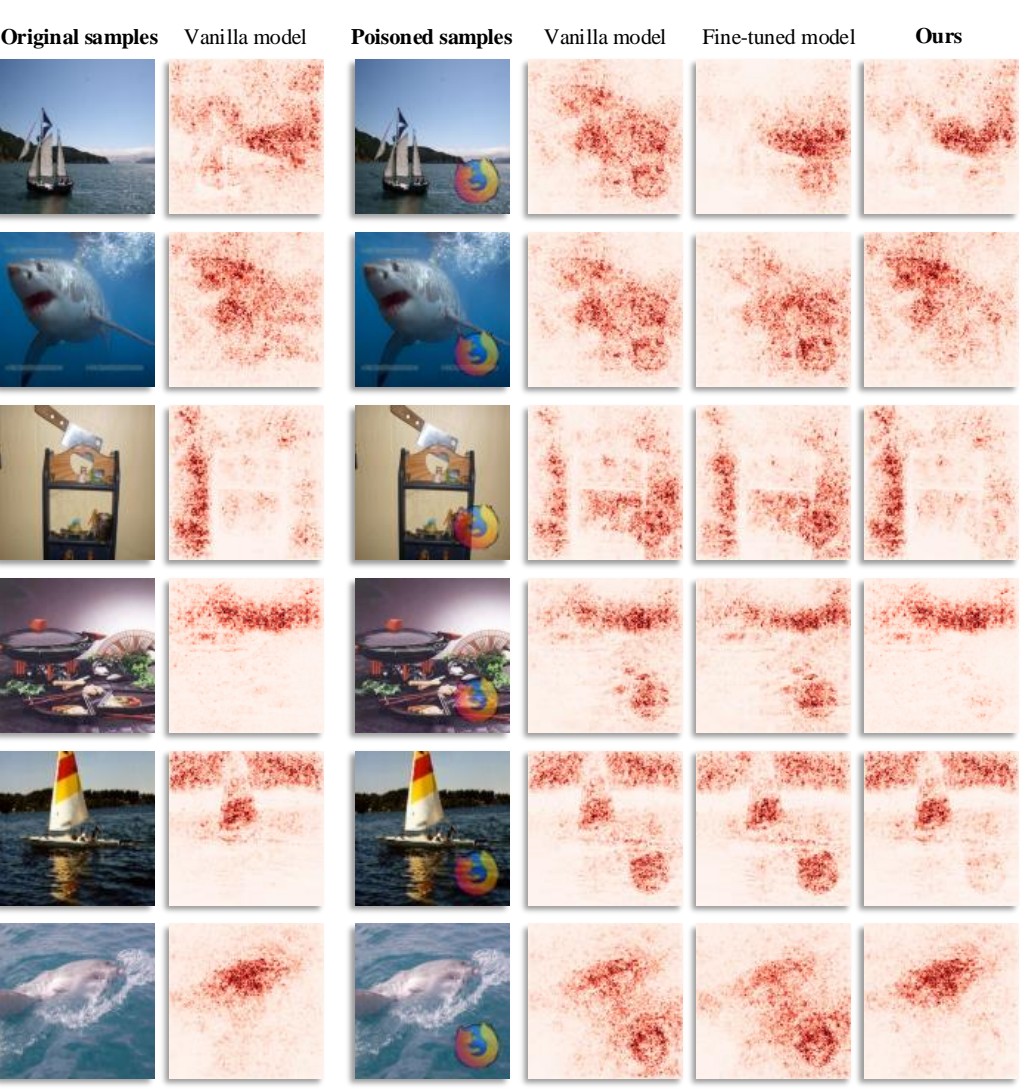

**Original samples**    Vanilla model    **Poisoned samples**    Vanilla model    Fine-tuned model    **Ours**

Figure 9: Attribution map comparisons on ImageNet among the vanilla model, fine-tuned model and dynamic edited model (Ours). When the model misclassifies poisoned samples containing triggers, our method effectively corrects this unreliable behavior, aligning the attribution maps with those derived from the original samples.

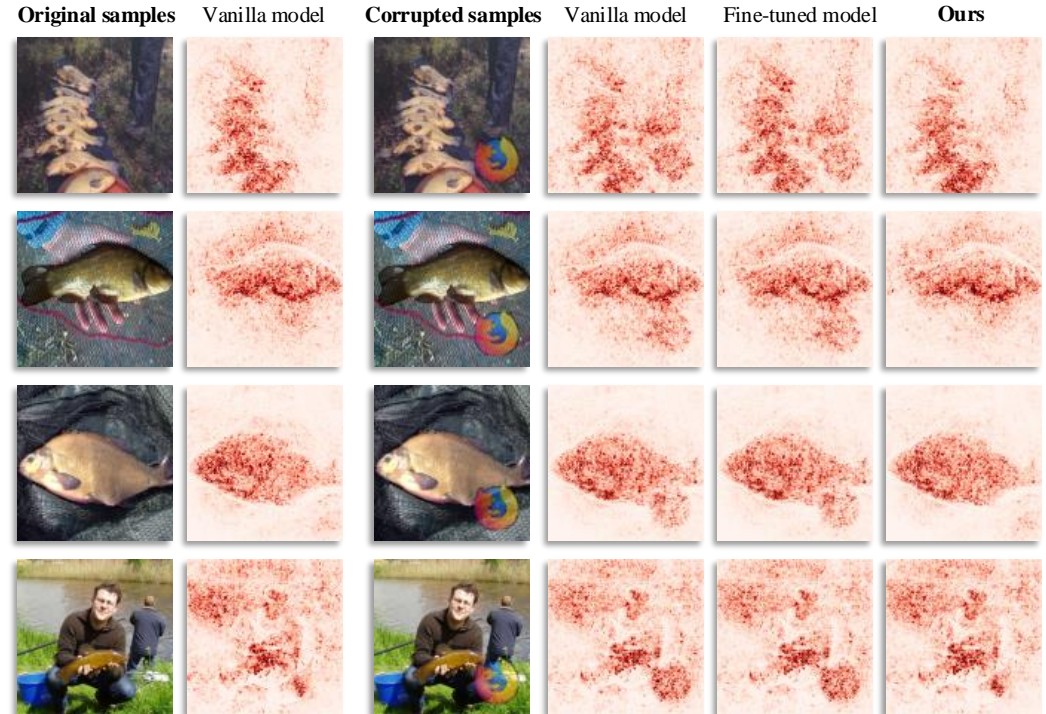

Figure 10: Comparisons of attribution maps on ImageNet among the vanilla model, fine-tuned model and dynamic edited model (Ours). Our method effectively mitigates the model's reliance on spurious correlated features, aligning the attribution maps with those derived from the original samples.

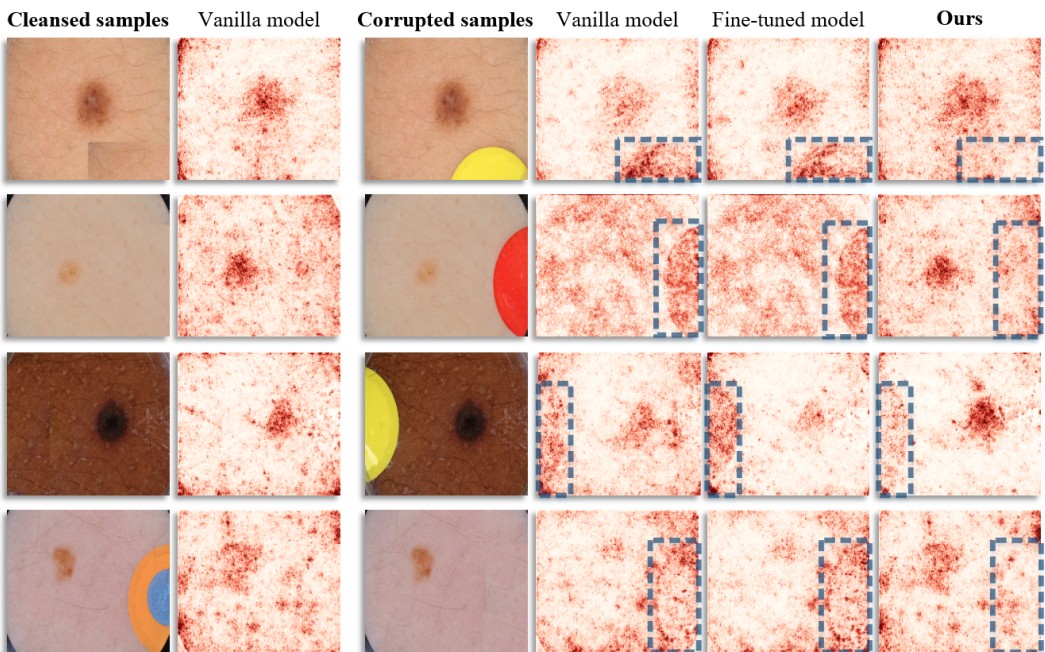

Figure 11: Comparisons of attribution maps on ISIC dataset among the vanilla model, fine-tuned model and dynamic edited model (Ours). When the model relies on the spurious feature to make predictions, our method effectively corrects this unreliable behavior, aligning the attribution maps with those derived from the original samples.

