# OpenReview forum: "Dynamic Model Editing to Rectify Unreliable Behavior in Neural Networks"
_ICLR.cc/2025/Conference — Submitted to ICLR 2025_

### Official Review · Reviewer_1hwe · 2024-10-29

**Soundness:** 2
**Presentation:** 2
**Contribution:** 2
**Rating:** 6
**Confidence:** 3

**Summary:**

This work considers the problem of spurious correlations learned by a model during training. It proposes the use of rank-1 editing as an approach to correct such model errors while preserving the model’s overall performance. This method is motivated by two challenges in the use of rank-1 editing. One is technical and involves finding rank-1 updates that do not interfere with other facets of model performance. The second is more specific to domain adaptation and involves the need for sufficient quantities of labeled data. The method first utilizes a feature attribution-based approach to locate the layer of the model where editing will yield the biggest improvement. Then it applies rank-1 editing to this layer to correct the spurious correlation in the model. The paper evaluates the approach on models to which a trojan has been injected and models that have learned spurious features related to patches (for both toy and real datasets). Experiments suggest that the approach strikes a good balance between correcting model behavior in specific instances without degrading overall accuracy too much.

**Strengths:**

**Promising research direction:** Domain adaptation is a serious issue in the application of machine learning to many domains. While the use of rank-1 editing has been proposed before in this setting, the reviewer believes that the suite of tools that editing offers have still not been fully exploited.

**Methodical experiments and results are strong:** Overall, the experimental section is well-written. While the reviewer has some issues with the overall scope of the experiments (see Weaknesses), those that were run seem to be fairly comprehensive and the results are well-described. In the settings where the method is deployed, it performs well relative to the other baselines that are explored.

**Weaknesses:**

**The problem that is addressed is narrow:** This work claims to explore domain adaptation, but it only looks at mitigating the presence of very obvious spurious correlations in the data (e.g., trigger patches). The challenges associated with real-world domain adaptation are far more subtle than this. It would have made the work stronger if it had investigated instances of domain adaptation where the differences between domains were more subtle. If the proposed method worked in such situations, it would be very notable. Alternatively, the paper could shift its language to focus more exclusively on spurious correlations.

**The challenges that motivate the method are not very clearly described and are never shown empirically to be issues:** The work describes two challenges that are meant to motivate the proposed approach. Overall, these are not very clearly described. Indeed, Challenge 2, which involves lack of data in domain adaptation settings, could be easily described outside of the mathematical formalism, but this is not done. Challenge 1 relates to the specific approach to rank-1 editing, specifically the failure of $k^*$ to be included in the statistics matrix $C$. This is stated as one of the fundamental challenges of using rank-1 editing for domain adaptation, but it is never explained why this is specifically a problem for domain adaptation and not a general issue. Finally, it would help make these challenges more meaningful if some empirical evidence was given to support their centrality.

**Repeated editing has been explored in the past:** To this reviewer’s understanding, the main contribution of the work is the use of feature attribution to locate a layer to edit, the modification of the existing rank-1 editing technique to mitigate an issue with the statistical matrix $C$, and the introduction of dynamic editing. The first and second are new to this reviewer’s knowledge (though the reviewer is not an expert in the breadth of what has been done in the editing space). The impact of repeated editing has been explored in detail in past works (e.g., [1]). It would be good to consider how the present paper fits into such studies.

### Nitpicks:
- Line 077: “Experimental evaluations highlight our method’s remarkable performance…” It is this reviewer’s opinion that the word ‘remarkable’ should be removed and that the paper should let the results speak for themselves.
- Line 043: The first sentence says that there are significant challenges to using rank-1 editing for domain adaptation. The second sentence says that actually rank-1 editing is well-suited to domain adaptation. What changed?

[1] Gupta, Akshat, Anurag Rao, and Gopala Anumanchipalli. "Model editing at scale leads to gradual and catastrophic forgetting." arXiv preprint arXiv:2401.07453 (2024).

**Questions:**

- Equation (1): What is $f(k^*;W)?
- Lemma 2: What does $x^* \rightarrow k^*$ mean? What is $\mathcal{X}$?

---

> ### Author Response · Authors · 2024-11-18
> **Response to Reviewer 1hwe**
>
> We sincerely thank the reviewer for their thoughtful feedback and for highlighting areas for improvement. Below, we address the specific concerns raised.
>
> > The problem that is addressed is narrow … claims to explore domain adaptation …
>
> We would like to clarify that our work is not about exploring domain adaptation, but rather correcting model unreliabilities such as caused by spurious correlation and neural Trojan. We will make this even more clear in the final paper.
>
> The primary focus of our paper is on addressing specific types of model unreliabilities, and we have chosen to frame our language around this focus. We will update the paper accordingly to ensure this distinction is clearer.
>
> > … challenges are not very clearly described and are never shown empirically …
>
> **Motivation and Scope.** Our work focuses on correcting unreliable model behavior by repurposing rank-one model editing for this task, rather than directly addressing the inherent challenges of applying rank-one model editing to domain adaptation. The identified challenges serve to illustrate critical limitations of rank-one editing in domain adaptation, which we sidestep in targeting a distinct task. The task shift makes it extremely hard to fairly compare performance across tasks, which does not make for convincing empirical evidence. Thus, we provide theoretical proofs (in App. A.1) to substantiate the identified challenges.
>
> **More on Empirical Evidence.** While directly evaluating the impact of challenges within domain adaptation is beyond the scope of our work, our results demonstrate the advantages of our method within model unreliability correction. Specifically, our approach effectively corrects model behavior with minimal samples while retaining high overall performance. These benefits contrast with applications of rank-one editing in domain adaptation, which often require numerous samples and lead to more significant overall performance degradation. These outcomes underscore the importance of sidestepping the identified challenges.
>
> **Improving Clarity.** To enhance clarity, we will further summarize the challenges in a more easily understandable way in our revision. For challenge 1, in domain adaptation, the model learns associative mappings that exclude $k^*$, as $k^*$ represents features that remain unaligned across domains. However, in reliability correction, $k^*$ corresponds to features aligned with the model's learned domain, necessitating its inclusion in $C$.
>
> > Repeated editing has been explored in the past …
>
> We thank the reviewer’s valuable suggestions and for bringing up the related work [1].  While repeated editing has been explored in [1], we believe our approach introduces key novelties. In contrast to [1], which focuses on the capacity of a fixed layer to be repeatedly edited, our method introduces dynamic layer identification, allowing the model to select layers for editing based on susceptibility. This dynamic selection extends the concept of repeated editing from a fixed layer to the entire model, thus providing greater flexibility and adaptability for further exploring the potential of the edited model. To the best of our knowledge, our work is the first to propose model editing specifically for correcting unreliable behaviors, as opposed to incorporating new associations in [1]. We will incorporate a discussion of this in the revision.
>
> > Line 043 & 077
>
> We appreciate the reviewer’s detailed reading. However, we believe there may have been a misinterpretation. In lines 043-046, we clearly state that rank-one model editing is well-suited for correcting a model’s unreliable behavior, not specifically for domain adaptation. The phrasing in our manuscript is precise on this point, but we will review the text to ensure that it is unequivocally clear and minimizes any potential for misreading.
> For line 077, to maintain an objective tone, we will revise the wording to allow the evaluation results to stand on their own merit.
>
> > Questions
>
> 1. In Equation (1), $f_l(k^*;W’)$ represents the mapping of a feature key $k^*$ by a layer $f_l$ with weights $W’$. We acknowledge the omission of the layer index $l$ and will correct this typo in the revision.
> 2. For Lemma2, the notation $x^* \rightarrow k^*$ refers to the mapping of an input $x^*$ to its corresponding feature map $k^*$ through the intermediate layers of the neural network. Additionally, $\mathcal{X}$ represents the input set of samples used to train the network. We will revise the explanation in the manuscript to make these definitions clearer.

---

> > ### Comment · Reviewer_1hwe · 2024-11-23
> > **Reply to authors**
> >
> > We would like to thank the authors for their clarifications. Given that the authors will make both the challenges and proposed scope of the paper more clear and given some of the extra experiments that were suggested by other reviewers and then run by the authors, I am raising my score to a 6. I like the ideas behind this paper and hope it finds a good home. That being said, this paper sits at the confluence of a lot of areas of machine learning (e.g., model editing, model robustness, etc.). There is thus a burden of contextualizing the work in relation to each of these areas. Having done this, I believe the work will be stronger.

---

> > > ### Author Response · Authors · 2024-11-23
> > >
> > > We sincerely thank Reviewer 1hwe for their thoughtful feedback and for raising their score. We appreciate your acknowledgment of the ideas behind our paper and your encouragement as we continue refining our work.

---

### Official Review · Reviewer_Nh8i · 2024-11-03

**Soundness:** 3
**Presentation:** 2
**Contribution:** 3
**Rating:** 5
**Confidence:** 3

**Summary:**

The authors apply model editing (very similar to ROME) for the task of removing neural trojans and mitigating reliance on spurious correlations (which in the paper, closely resemble neural trojans) for image classifiers. About half the paper is dedicated to i. explaining ROME, ii. describing some challenges in applying editing to the desired settings, and iii. detailing their modified approach. Their method consists of identifying the best layer to edit by comparing attributions (i.e. via Integrated-Gradients) for clean and poisoned samples, and then "dynamically" applying model editing, which I believe (but am unsure) means making repeated edits until the edited model's overall performance does not deviate much from the original model's overall performance.

Experiments are conducted using a simple neural trojan and spurious feature for CIFAR10 and ImageNet classification. The proposed method successfully reduces to the attack success rate of the neural trojan to nearly zero without compromising overall accuracy. It also removes any bias to the spurious feature (measured as the increase in performance when the spurious feature is present). Baselines include fine-tuning and methods called P(or A)-ClArC, and are surpassed. Similar results are attained on a more realistic setting of skin lesion classification.

**Strengths:**

The method seems to work well! ASR is dropped to nearly zero without compromising accuracy, and bias toward spurious features are removed without affecting accuracy.

The result could be an important result for the mechanistic interpretability community (which is currently garnering lots of attention) as it shows editing techniques can be applied to a second modality and to alleviate existing concerns.

I really liked that a realistic setting was also considered, and that the method seemed to work well in this case too.

**Weaknesses:**

**Clarity**: I personally found this paper hard to read. I did not find Sec 4 to be well integrated to the paper. Paragraph 207-215 feels like it should be very important, but it was not clear to me how your method changed to sidestep concerns and if there was empirical evidence showing that this methodological change was truly responsible for improved performance. The paper would have benefitted more from taking more time to clearly explain the experiments, imo.

**Novelty over ROME**: Perhaps this relates to the above point, but it is unclear to me exactly how this differs from ROME, which as I understand it, also involves localizing and editing. The iterative nature (which authors term 'dynamic') is perhaps new, but it seems to only marginally improve performance over static editing. Similarly, the comparison to Santurkar's editing work was a bit lackluster (is the main difference that they choose the penultimate layer by default?)

I am **unsure if this method would work for more realistic spurious correlations**, which would not have a single fixed appearance, as is the case for the spurious features studied (even for the skin lesions, the spurious patches are quite consistent in their appearance). Even a simple benchmark like Waterbirds is not studied (I personally think even Waterbirds is too simple, but it is very established and having a result on it would greatly improve the paper's claim about spurious features).

Summary: I am borderline on this paper. The clarity issues are somewhat significant for me, but the experimental results are strong and the impact of showing editing is effective for vision models would be impactful. I am curious if other reviewers also had difficulty reading the paper -- if it is just me, I'd raise my score. Novelty issues are not as big for me, but I think the paper would be strengthened if the exact differences between this and the most similar related methods are clearly and concisely articulated.

**Questions:**

What is the difference between your method and ROME, aside from (i) applying it to image classifiers (ii) editing iteratively (or 'dynamically'), and (iii) the inclusion of corrupted samples in training (pls correct me if I am interpreting this wrong -- see next question)?

Does this mean you train on the corrupted samples? What if your model has already been trained? Also, don't the corrupted samples need to be included in training to begin with, so that the attack is successful? This part is unclear to me.
> L207: "Our proposed process of model editing to correct unreliable behaviors involves integrating both original samples x and their corrupted counterpart x˜ into the training procedure"

More minor:
Don't these sentences contradict each other? Maybe you need a name for your method to distinguish it from the standard ROME (which you say doesn't work out of the box).
> L43: "we formally pinpoint two key challenges when applying rank-one editing to domain adaptation, which inevitably lead to diminished model performance and necessitate labor-intensive data preparation (details in § 4.1). Next, we establish that rank-one model editing is
well-suited for correcting model’s unreliable behavior as it intrinsically sidesteps these challenges"

Some questions for figure 3:
- Should it be key k*? Instead of key v*? (step 3 panel)
- Yellow arrow is pointing wrong way and should be under 'attribution flow'?

What patterns? I think this is important -- it is a lot easier to edit out a reliance on a spurious pattern that does not vary much in its appearance (like in the Trojan case) than it is to edit real spurious features (e.g. to backgrounds).
Update: I see in the appendix the patterns added are the same as the Trojans -- the only difference in the settings are that the added patterns flip the label in the Trojan case, while they do not in the spurious case. I think this makes your spurious correlations setting unrealistic.
> L410: "we pollute a proportion of samples of class y by attaching patterns to create spurious samples"

Suggestion for table 4: highlight the accuracy on the samples without the spurious correlation (is this what you mean by clean?) or performance drop for these samples instead of showing the performance for samples with the correlation. It reads a little cleaner to see your method improves accuracy, and better highlights the cost of relying on spurious features (i.e. performance is worse when the feature is absent + correlation is broken).

---

> ### Author Response · Authors · 2024-11-18
> **Response to Reviewer Nh8i (1/2)**
>
> We thank the reviewer for their recognition and constructive suggestions. We provide our response to your comments below.
>
> > Clarity ...
>
> We do want to note that Reviewer RTf7 explicitly notes that the paper is “well written” and mentions “this is an enjoyable paper to read”.
>
> **On sidesteping challenges.** To clarify how our method sidesteps the challenges identified, we provide a rigorous discussion: For a susceptible model, the training process integrates both clean samples and their corresponding corrupted counterparts. This integration ensures: $C=KK^T, K=[k_1, k_2, …, k^*], V=[v_1, v_2, …, v^*]$. This eliminates the residual $r$ such that $C^{-1}k^*\in span(K)$. Thus, the unchanged key-value associations preserve model performance, sidestepping Challenge 1.
> By incorporating \{$x,\tilde{x}$\} $\in \mathcal{X}$ in training, the model ensures $||k^*-f(x^*;W_D)||->0$ as $x^*\in \mathcal{X}$, when $|\mathcal{X}|>>0$ is not available. It mitigates the need for extensive data preparation, sidestepping Challenge 2.
> Please note that Paragraph 207-215 is a part of Sec 4.2 which focuses on handling the challenges highlighted in Sec 4.1. The challenges in 4.1 are for the ‘domain adaption’ problem for which rank-one editing has been used previously (while facing those challenges). We propose the first use of rank-one editing for handling backdoor and spurious correlation. We do not let those challenges limit our method by sidestepping them. Section 4 is carefully organized to provide this clear picture.
>
> **On providing empirical evidence.** Our work focuses on repurposing rank-one model editing to rectify unreliable model behavior. Our objective is not about addressing the challenges of rank-one editinging domain adaptation. We deal with a different problem of unreliable model behavior. The challenges we identify in domain adaptation identify the limitations of rank-one editing, which our method sidesteps for our task. A fair experimental comparison across different tasks is nontrivial. To address concerns regarding evidence, we emphasize that our theoretical discussion validates the identified challenges. Nonetheless, our results demonstrate the advantages of our method in correcting model unreliability. Specifically, our approach achieves effective behavior correction with minimal samples while maintaining high overall performance, contrasting with domain adaptation applications that typically require more samples and suffer greater performance degradation. We will further clarify this in the revision.
>
> > Novelty over ROME …
> > Question 1
>
> We thank the reviewer for recognizing our contributions. Below, we clarify the additional differences, and articulate the unique contributions of our approach:
>
> **Objective focus.** Unlike ROME and related works that focus primarily on domain adaptation, our method is aimed at correcting model unreliability, specifically addressing issues such as spurious correlations and neural Trojans. This distinct focus enables us to sidestep challenges typically encountered in model editing and facilitates effective correction of undesired behaviors.
>
> **Layer localization technique.** We introduce a novel attribution-based layer localization technique to identify layers contributing to unreliable behavior. In contrast to methods editing with a fixed layer (e.g., the penultimate layer in Santurkar et al.), our technique allows for flexible layer selection across the model. This not only enhances editing effectiveness but also broadens the model’s editing capacity from a single layer to the entire model.
>
> **Efficiency with Minimal Data.** Our approach achieves remarkable data efficiency, requiring as few as a single cleansed sample while retaining the model overall performance. In contrast, ROME often demands extensive data preparation and leads to greater performance degradation.

---

> > ### Author Response · Authors · 2024-11-18
> > **Response to Reviewer Nh8i (2/2)**
> >
> > > … work for more realistic spurious correlations …
> >
> > We would like to clarify two points regarding the spurious correlations studied in our work. The spurious patches in the ISIC dataset are indeed challenging. As illustrated in Fig. 7, the spurious patches exhibit diverse patterns and locations, with no two patches being identical. This variability makes the task more complex than might initially be assumed.
> >
> > On the other hand, while the trigger patterns in the blend attack are consistent by design, this choice strikes a critical balance between their stealthiness and their influence on the model’s predictions. The blend attack is a strong baseline because its consistent patterns are highly effective in biasing the output prediction, allowing us to rigorously evaluate our method under well-established adversarial settings.
> >
> > We also appreciate the suggestion to include results on Waterbirds and have provided them in the table below. These results demonstrate that our method consistently corrects the model’s unreliabilities, further substantiating its robustness to spurious correlations in diverse datasets.
> >
> > | Method | Worst-Group Accu. | Overall Accu. |
> > -|-|-
> > | Vanilla Model | 62.90 | 87.70 |
> > | Group DRO | 63.60 | 87.60 |
> > | Fine-tuned Model (n=10) | 63.12 | 86.50 |
> > | Edited Model (n=10) | 66.84 | 87.64 |
> > | Dyn. Edited Model (n=10) | 69.18 | 87.68 |
> >
> >
> > > Questions: Does this mean you train on the corrupted samples? …
> >
> > To clarify: the corrupted samples should be included during the training process to ensure the success of the attack. Our proposed model editing technique leverages both corrupted samples $\tilde{x}$ and their corresponding clean samples $x$ in the training process. This distinguishes our approach from methods primarily targeting domain adaptation, as our focus is on rectifying unreliable model behaviors rather than adapting to a new domain. We acknowledge that the phrasing in L207 could be clearer, and we will revise it to: “Our proposed model editing process integrates both the original samples x and their corrupted counterparts x˜ into the training procedure.”
> >
> > > Questions: More minor ...
> >
> > Thank you for the opportunity to clarify. We will revise this section for clarity, emphasizing that the shift in the task—from domain adaptation to correcting unreliable model behavior—leads to the observed effectiveness of rank-one model editing in overcoming the identified challenges.
> >
> > > Questions: figure 3 ...
> >
> > Thank you for pointing out these details. We will correct the key to $k^*$ as suggested. Regarding the yellow arrow, we will reverse its direction and reposition it under the "attribution flow" label to improve clarity.
> >
> > > Questions: ... patterns ...
> >
> > We appreciate your point about the ease of editing spurious patterns in the Trojan case. In addition to evaluating spurious correlations with trigger patterns, we also conduct experiments on the ISIC dataset. To further address your concern, we have added an additional experiment on the Waterbirds dataset, which presents a more realistic scenario. We hope these clarifications along with the additional experiment address  your concerns.
> >
> > > Suggestion for table 4 ...
> >
> > Yes, by "Clean" we are referring to the samples without the spurious correlation. We will update the table and clarify this in the revised version to better highlight the performance improvements when the spurious feature is absent and to emphasize the cost of relying on spurious correlations.

---

> > > ### Comment · Reviewer_Nh8i · 2024-11-24
> > >
> > > thank you for your work during this rebuttal period! Unfortunately I still have some questions / concerns:
> > >
> > > 1. Some of the numbers in your Waterbirds experiment seem oddly low... Group-DRO is usually reported to have around 90% worst group accuracy (see the original DRO paper, along with Kirichenko et al's last layer retraining paper). Do you have an explanation for this large discrepancy?
> > >
> > > 2. What does a 'cleansed' example look like in the Waterbirds case? And what about in general? Do you always need to have knowledge of the spurious feature, and a way to inpaint or remove it?
> > >
> > > Some of the arguments in the rebuttal also haven't quite landed with me -- mainly the idea that ROME is designed only for domain adaption and that 'rectifying unreliable behavior' is significantly different from that. Wasn't ROME originally designed for correcting factual mistakes, and wouldn't this fall closer to 'rectifying unreliable behavior' than domain adaptation?
> > >
> > > In general, I think I am a more critical reviewer than most, so if you have sufficient answers to these questions, I'll raise my score to a 6 in an attempt to self-normalize, even though I may still have reservations about the work.

---

> > > > ### Author Response · Authors · 2024-11-25
> > > > **Response to Follow-Up Questions**
> > > >
> > > > We sincerely thank the reviewer for their feedback and for continuing to engage with our work. Below, we address the remaining concerns.
> > > >
> > > > 1. Our reported results are fully reproducible using the Group-DRO implementation provided in the original paper. The discrepancy arises because our experiments use a ResNet-34 backbone with standard regularization, whereas the original DRO paper [1] reports a worst-group accuracy of 76.9% for ResNet-50 under a similar setup. This setup is now described in our revisions. The ~90% accuracy cited by the reviewer typically combines Group-DRO with group adjustment and a strong $L_2$​ penalty, resulting in higher performance. These additional techniques were not applied in our experiments, as we intended to compare basic training strategies. For context, _our method requires only minutes for editing, whereas retraining strategies often demand several hours and a full assessment of training data_. Additionally, the spurious correlations in the Waterbirds dataset stem largely from the imbalanced distribution of samples across categories. Our method is not specifically aimed at distributionally robust optimization.
> > > >
> > > > 2. In the Waterbirds dataset, cleansed samples are those where the true feature (e.g., a waterbird) is presented alongside the correct/non-spurious background (i.e., water background). Conversely, spurious samples depict waterbirds with a land background. We created only 10 cleansed samples for our experiments, which is a very small fraction of the training set. Compared to training strategies, which also presuppose prior knowledge of spurious features and require full access to the dataset, our method has the advantage of superior efficiency and adaptability with minimal data. For scenarios where dataset access or manual cleansing is not feasible, future work could explore using generative models to aid in generating cleansed samples.
> > > >
> > > > 3. We apologize for any miscommunication and thank the reviewer for prompting clarification. Rank-One Model Editing (ROME) was initially proposed for generative models [2] and later extended to predictive classification models for tasks such as domain adaptation [3]. Recent work has also explored ROME for editing behavior in generative language models (e.g., GPT) [4, 5]. Our work differs fundamentally in both scope and focus. While ROME applications address domain adaptation or factual association adjustments, our method directly targets unreliable behavior in predictive models. Crucially, _our contributions include a novel layer localization technique, which pinpoints the layers responsible for unreliable behavior before applying rank-one edits_. This addition distinguishes our method from existing ROME-based techniques and enhances its effectiveness for rectifying model behavior in predictive tasks.
> > > > We hope these clarifications address the reviewer’s concerns. We will further clarify these distinctions in our revisions. We appreciate the reviewer’s critical evaluation and are committed to improving it. We will be happy to clarify any subsequent comments the reviewer might have. Thank you!
> > > >
> > > > [1] Sagawa, Shiori, et al. "Distributionally robust neural networks for group shifts: On the importance of regularization for worst-case generalization." ICLR (2019).
> > > >
> > > > [2] Bau, David, et al. "Rewriting a deep generative model." ECCV (2020).
> > > >
> > > > [3] Raunak, Vikas, and Arul Menezes. "Rank-one editing of encoder-decoder models." arXiv preprint (2022).
> > > >
> > > > [4] Meng, Kevin, et al. "Locating and editing factual associations in GPT." NeurIPS (2022).
> > > >
> > > > [5] Santurkar, Shibani, et al. "Editing a classifier by rewriting its prediction rules." NeurIPS (2021).

---

> ### Comment · Reviewer_Nh8i · 2024-11-29
>
> tldr for AC: I still have reservations about this paper’s scope, novelty, and presentation. However, the idea of editing models to improve reliability has long been of interest to the community, and this seemingly positive result could spur further research. **I’d advise the AC to accept this paper if they find the overall application of editing (to fix some backdoors and spurious correlations in image classifiers) sufficiently compelling to outweigh my listed reservations**. I will be maintaining my score though.
>
> ----
> longer:
>
> **Discussion of rebuttals**
> I seem to share the same thoughts as reviewer 1hwe – while the rebuttal experiment assuaged their concerns, it did not mine, as I don’t understand why the strongest version of DRO was not used. This leads to questionable results, where DRO oddly offers almost no gain. Thus, my concerns that the proposed method wouldn’t work for most spurious correlation cases linger, especially given the need for manual cleansing and that Waterbirds itself is still rather simple. If I could, I’d raise my score by 0.5 to acknowledge the effort; since reviewer 1hwe already increased their score by 1, I will keep mine as is.
>
> I also found the response to reviewer RTF7’s critique on the simplicity of the blend attack unsatisfactory. The authors argue the blend attack is strong, but I’d argue it is easier to thwart via simple detection, and also probably easier to edit out than other attacks. As mentioned, the current work assumes knowledge of the backdoor or spurious correlation to edit -- this is also a noteworthy limitation.
>
> **Overall pros and cons**
>
> Pros:
> - Good application of editing (fixing models with backdoors or spurious correlations)
> - Seems to work in the selected settings, including a real world spurious feature for skin cancer detection.
> - Seemingly new editing layer localization technique
>
> Cons:
> - Generally, it feels like the paper is oversold
>     - Novelty over ROME still feels slim to me, even after the authors’ rebuttal.
>     - The ‘rectifying unreliable behavior’ in actuality corresponds only to fixing (i) backdoors/trojaned models and (ii) spurious correlations, with the spurious correlations implemented in a way that is extremely similar to the trojans.
>     - Also, the ‘models’ in question are just simple image classifiers.
> - The key novelty – the layer localization technique – is not adequately studied / compared to baselines. The ‘dynamic’ part of the method seems to only marginally improve performance.
>
> **Feedback going forward**
>
> - I would reorganize this paper to focus on the novel elements. I did not find the theoretical discussion particularly insightful, and instead would have appreciated a deeper dive on the layer localization technique: which attribution technique is best? Is performance stable over different archs? Any explanation as to why different layers are selected, based on the trigger or dataset?
> - I’d expand experiments to test different kinds of triggers and settings. You can probably extend beyond image classifiers easily as well.
> - I’d avoid upselling the paper – mitigating backdoors is already interesting enough! A title change could be appropriate as well.
>
> ----
> **Final comment**: I thank the authors for their work on the paper and rebuttal. I think they are working on an important problem and I encourage them to see this paper through. This paper is definitely ready for additional feedback from Workshops, and when expanded, I believe will be impactful in its full form. I wish the authors the best of luck.

---

> ### Author Response · Authors · 2024-12-02
> **Response to follow-up quesionts (1/2)**
>
> We sincerely thank the reviewer for their feedback. We emphasize that Reviewer 1hwe was satisfied with our response, requesting no further clarification. Hence, our response did not go any further than satisfying the reviewer. It is a bit unfair to give a lower score based on the concerns that have already been addressed. Appreciating Nh8i’s reservations, we provide further clarification and hope that the reviewer re-evaluates our work based on the added information.
>
>
> ### For Rebuttals
>
> **1. Comparison with Group GRO.**
>
> We would like to clarify that our method addresses a different challenge compared to Group DRO. Group DRO is a training strategy specifically designed for distributionally robust optimization. Spurious correlations in datasets such as Waterbirds and CelebA are often a result of imbalanced data, which Group DRO mitigates in model training. In contrast, our approach focuses on post-training correction. This distinction is important as our method aims to rectify the behavior of established models, and its effectiveness also depends on the model's ability to recognize the true features.
>
> To address the concern about the performance of Group DRO in our experiments, we conducted additional tests where we applied Group DRO with stronger L2 regularization and group adjustments. As shown in Table 10, our method further enhances a model trained with Group DRO, achieving higher performance with a significantly reduced computational cost.
>
> Table 10. Experiments on Waterbirds.
> | Method | Worst-Group Accu. | Overall Accu. | Time |
> -|-|-|-
> | Vanilla Model | 62.9 | 87.7 | **545 mins** |
> | Group DRO + L2 + Group Adjustment | 87.4 | 92.3 | **649 mins** |
> | Edited Model (Group DRO) (n=10) | 88.9 | 92.5 | **8 mins** |
> | Dyn. Edited Model (Group DRO) (n=10) | 90.7 | 92.3 | **12 mins** |
>
>
> In Table 11, we also conducted experiments on the CelebA dataset to demonstrate the generalizability of our method. Using ResNet-34, we corrected spurious correlations related to gender and improved performance while keeping computational costs low. These experiments show that our method can achieve competitive results.
>
> Table 11. Experiments on CelebA.
> | Method | Worst-Group Accu. | Overall Accu. | Time |
> -|-|-|-
> | Vanilla Model | 49.9 | 95.0 | **273 mins** |
> | Group DRO | 59.4 | 94.9 | **278 mins** |
> | Group DRO + L2 + Group Adjustment | 85.3 | 94.9 | **324 mins** |
> | Edited Model (n=10) | 72.9 | 93.3 | **9 mins** |
> | Dyn. Edited Model (n=10) | 75.1 | 94.1 | **12 mins** |
>
>
>
> **2. Concerns on the simplicity of Blend attack to detect and defense.**
>
> To address the concern about the visible trigger patterns, we have conducted additional experiments **using the ISSBA method (Li et al., ICCV’ 2021), an invisible backdoor attack characterized by imperceptible perturbations as triggers**. These invisible triggers are more challenging to detect.
>
> In Tables 13 and 14, We evaluated ResNet-18 models trained on CIFAR-10 and GTSRB datasets. The results demonstrate that our proposed method is not limited to visible triggers and performs effectively even under invisible attack scenarios. Importantly, for ISSBA, our approach does not require knowledge of the exact trigger pattern but only clean samples. These results also align with the rationale for our methodology as detailed in Appendix 3.2.
>
> Table 13. Performance comparison of defending against the ISSBA backdoor attack on CIFAR-10.
> | Method | Overall Accuracy | Attack Success Rate |
> -|-|-
> | Benign Model | 93.8 | 100.0 |
> | Finetuned Model (n=10) | 91.6 | 26.4 |
> | Edited Model (n=10) | 92.6 | 3.5 |
> | Dyn. Edited Model (n=10) | 93.4 | 0.6 |
>
> Table 14. Performance comparison of defending against the ISSBA backdoor attack on GTSRB.
> | Method | Overall Accuracy | Attack Success Rate |
> -|-|-
> | Benign Model | 97.2 | 100.0 |
> | Finetuned Model (n=10) | 95.9 | 39.2 |
> | Edited Model (n=10) | 96.9 | 5.7 |
> | Dyn. Edited Model (n=10) | 96.3 | 1.4 |
>
>
> **3. Prior knowledge of backdoor or spurious correlation.**
>
> We would like to clarify that our work significantly mitigates the reliance on detailed knowledge of backdoors or spurious correlations. Specifically, we **achieve reduced reliance on cleansed samples** requiring as few as one pair of samples, and **eliminate the need for precise trigger information** by enabling image-level corrections. In contrast, existing post-hoc techniques for correcting model behavior often demand a full dataset assessment or precise trigger information. **We believe our approach strikes an excellent trade-off between minimal knowledge requirements and robust performance compared to related methods. For a post-training method like ours, it is impractical to require performance on par with training-based methods or to expect it to operate with no prior knowledge as the model attacker.**

---

> ### Author Response · Authors · 2024-12-02
> **Response to follow-up quesionts (2/2)**
>
> ### For Cons
>
> **1. Oversold.**
>
> We respect the reviewer’s perspective on the novelty. However, we have clearly defined the scope of "unreliable behavior" in the paper, with a detailed discussion in the related works section. Regarding spurious correlations, in addition to the ISIC dataset, we now include experiments on both Waterbirds and CelebA. We believe that referring to spurious correlations and backdoors as forms of unreliable behavior is both valid and consistent with the objectives outlined in our paper.
>
>
> **2. The improvement of the layer localization technique.**
>
> We must highlight that **the improvement brought by the layer localization technique is significant**.
>
> The layer localization technique is central to our method and is motivated by a critical observation: editing different layers yields distinctive performance outcomes. This is demonstrated in **Figures 2 and 8**, where we identify a serious limitation in existing methods that focus on editing only the last layer, often leading to suboptimal results or performance degradation.
>
> To ensure fairness, we avoid direct comparisons with methods constrained by this limitation in the main text. However, for completeness, we supplement these comparisons in **Tables 6 and 7** (Appendix A.6), which clearly demonstrate the effectiveness of layer localization in achieving robust performance. For ease of reviewing, we briefly conclude the main results in Tables 15 and 16.
>
> Table 15. Attack Success Rate comparison of defending against the backdoor attack on Trojaned models trained with the Phoenix logo on CIFAR-10 and ImageNet.
> | Method | CIFAR-10 | ImageNet |
> -|-|-
> | Edited Model (n=1) | 4.49 | 15.24 |
> | Dyn. Edited Model (n=1) | 0.65 | 6.15 |
>
> Table 16. Errorously increased accuracy for spurious features on models trained with the Phoenix logo on CIFAR-10 and ImageNet.
> | Method | CIFAR-10 | ImageNet |
> -|-|-
> | Edited Model (n=1) | +4.13 | +11.95 |
> | Dyn. Edited Model (n=1) | +1.50 | +6.93 |
>
>
> In addition to the aforementioned experiments, we have designed a variant of the Blend attack that fixes the penultimate layer to manipulate the model's prediction using triggers. The results, presented in Table 17, demonstrate that the default setup used in existing model editing work is vulnerable to this simply modified attack. In contrast, our layer localization method effectively mitigates this vulnerability, showcasing its advantages in enhancing model robustness.
>
> Table 17. Comparison of attack success rate on models trained under a Blend attack with the fixed penultimate layer on CIFAR-10 and ImageNet.
> | Method | CIFAR-10 | ImageNet |
> -|-|-
> | Benign Model (Fix the penultimate layer) | 100.0 | 85.2 |
> | Edited Model (n=1) | 97.3 | 84.7 |
> | Dyn. Edited Model (n=1) | 0.9 | 1.8 |
>
>
> ### For Going Forward
>
> We thank the reviewer for the valuable suggestions to improve our work. **We would like to clarify that many of the suggestions have already been addressed or are incorporated in the current version.** The concerns, particularly regarding attribution methods and the mitigation of backdoor Trojans, are important and have been considered with respect to readers from diverse backgrounds.
>
> **1. Alternative attribution methods.**
>
> The selection of the path attribution technique is based on its ability to measure changes between samples while satisfying the Completeness axiom, ensuring reliability in identifying critical layers. **Other methods, including gradient-based or perturbation-based techniques, do not meet these requirements for measuring attribution change. Thus, experiments for testing different attribution methods are not feasible in this context.**
>
> To show the stability of our layer localization technique across architectures, we have provided results in Table 9 (Appendix A.8), which demonstrate consistent performance across different model architectures.
>
> **2. To test different kinds of triggers and settings.**
>
> We have already tested **different trigger patterns**, as shown in Tables 1, 6, and 7. Additionally, in Tables 2 and 3, we analyze the generalizability of our method on models Trojaned with **triggers of varying visibility and locations**. We have also included experiments on **invisible trigger patterns**, offering a comprehensive discussion on trigger variations and settings.
>
> **3. A title change.**
>
> We thank the reviewer for recognizing our contribution.  Whereas we maintain that the title reflects the content well, we are open to adjusting it. The reviewer can suggest an appropriate title.

---

### Official Review · Reviewer_xpVb · 2024-11-04

**Soundness:** 2
**Presentation:** 2
**Contribution:** 2
**Rating:** 3
**Confidence:** 5

**Summary:**

Summary: This paper proposes a novel method for editing unreliable models, drawing inspiration from rank-one editing techniques. The authors demonstrate the effectiveness of their approach through experiments focused on backdoor attacks and spurious correlations.

**Strengths:**

Strengths:
1. The method's approach of retaining original samples alongside corresponding corrupted samples effectively addresses issues related to performance degradation and data volume constraints.
2. Algorithm 1 employs a clever strategy for dynamically editing the model using attribution methods by appropriately setting thresholds for $\delta$ and $\epsilon$.
3. The experimental section considers critical issues such as backdoor attacks and spurious correlations, providing an analysis of the method on the real-world ISIC dataset, which showcases the method's extensive effectiveness.

**Weaknesses:**

Weaknesses:
1. The proportion of the backdoor subset within the training set is not clearly specified. To better evaluate the robustness of the proposed method, I recommend varying the proportion of backdoor data and comparing results across different configurations.
2. While the experiments demonstrate strong performance on image data, additional experiments in other domains, such as sequence recognition, would help establish the method's scalability and versatility, highlighting its broader applicability.
3. The paper lacks a comparative analysis of the time complexity of the proposed method relative to existing techniques. Including this analysis would offer valuable insights into the method's efficiency and practical feasibility.

**Questions:**

Question: The model editing process, as illustrated in Figure 3, involves clean samples and their corresponding corrupted samples. How would one edit a model trained on a dataset containing poison and Trojan horses when the original training dataset or clean samples are unavailable?

---

> ### Author Response · Authors · 2024-11-18
> **Response to Reviewer xpVb**
>
> We appreciate the reviewer’s feedback and provide our detailed response below.
>
> > … varing the proportion of backdoor data and comparing results across different configurations.
>
> We would like to clarify that the detailed experimental setup, including the poisoning rate, is already provided in Appx 3.1 and referenced in Line 331 of the manuscript. Specifically, we poison $0.1$\% of training samples $x$ with label $y\neq y^*$ to embed the backdoor trigger on ImageNet. For CIFAR-10, we set the poisoning rate of $1$\%. These poisoning rates are consistent with standard configurations in backdoor attack research, ensuring a realistic balance between attack stealth and effectiveness. We believe these configurations sufficiently demonstrate the robustness of our proposed method, and varying the attack proportions is unlikely to impact the conclusions of our results.
>
> > … experiments in other domains …
>
> We appreciate the reviewer’s suggestion to further demonstrate the method's applicability. In our submission, we intentionally focused on image datasets to establish the core effectiveness of the method. However, we agree that highlighting the scalability and versatility of our approach is valuable. To address this, we will clarify in the Introduction that the current scope of this paper is limited to image-based experiments. We will also emphasize that our method is inherently modality-agnostic, as the underlying procedures are generic and can be readily extended to other data types, such as sequences or graphs. This generalizability stems from the design of our approach, which does not rely on image-specific assumptions. We recognize this as an important avenue for future research and will include a statement to this effect in the revision.
>
> > … analysis of the time complexity …
>
> We thank the reviewer’s suggestions regarding time complexity analysis. Below, we detail the time complexity of our method and highlight its efficiency.
> Let $L$ denote the number of layers in the model, and $p$ the number of key-value pairs used in the editing process. The time complexity of our approach is primarily determined by two key components:
> 1. Attribution computation: This step involves a forward and backward pass through the model, scaling as $O(A_L)$, where $A_L$ is the costs of the forward and backward passes across all layers.
> 2. Rank-one editing: The cost of inverting the static matrix $C \in \mathbb{R}^{p \times p}$ is $O(p^3)$. This inversion is computationally lightweight in our method since we use only a few cleansed samples, resulting in small $p$.
>
> Given that the editing process iterates until convergence, let $T$ denote the number of iterations required. The total time complexity of our algorithm is $O(T\cdot (A_L+p^3))$. Our algorithm’s efficiency arises from two factors:
> 1. Small $p$: Since $p^3$ depends on the number of key-value pairs (determined by the cleansed samples), this term remains small due to our use of minimal samples.
> 2. Few iterations T: In practice, $T$ is smaller than the number of layer $L$ in the model, ensuring the overall time complexity remains low.
>
> In comparison to existing model editing techniques that rely on larger cleansed samples and model retraining, our approach achieves competitive computational efficiency, which is evident as the total complexity remains bounded by $O(T\cdot (A_L+p^3)) \leq O(L\cdot (A_L+p^3))$.
>
> > … when the original training dataset or clean samples are unavailable …
>
> As a model defender, clean samples are always accessible in practice, particularly when there is some knowledge about the domain or application. Additionally, the presence of corrupted samples can typically be inferred through existing anomaly detection or unreliable behavior detection techniques, which, while not the direct focus of our method, are well-established in related research.
>
> Even in scenarios where the original training dataset is unavailable, our method offers strong capacity for practical use due to the following advantages:
>  - **Minimal Cleansed Data Requirement.** Our method demonstrates effectiveness with minimal data, requiring as few as a single pair of clean and corrupted samples to perform effective model editing.
>  - **Image-Level Editing.** Unlike methods that demand precise identification of unreliable patterns, our approach operates at the image level, eliminating the need for precise localization of corruption within samples.
>
> Additionally, we provide further discussion on the identification of clean and corrupted samples in Appendix 8, offering insights into handling such scenarios effectively.

---

> ### Comment · Reviewer_xpVb · 2024-11-26
>
> The definition of "UNRELIABLE BEHAVIOR" is unclear. It is quite confusing why the proposed rank-1 approach can alleviate the challenges. There is no evidence supporting the claim that rank-one model editing can "correct the model’s unreliable behavior on corrupt or spurious inputs." Additionally, the most recent reference for the study of UNRELIABLE BEHAVIOR dates back to 2019.
>
> The key value optimization presented in Eq. (1) is a well-established formula developed by others. Therefore, the authors' contribution should primarily focus on Eq. (2). This clearly requires a gradient disagreement for estimating the function disagreement and necessitates a closed form for f.
>
> I am uncertain how this operates within neural networks. Essentially, it is not straightforward to define a feature mapping function for each layer. Relying solely on the loss function in the final iteration may lead to issues. For Eq.~(2), updating
> $x$  requires a trustworthy direction and appropriate step sizes. It is unclear why $\hat x$  is updated using  $x$ as a given teacher, as this could result in significant problems if the update is low-confidence.
>
> Overall, the presentation raises a significant number of concerns. It would be beneficial to reorganize the work, as it should be more self-explanatory.  The current statement comes across as a self-promotional piece of art, although there may be useful ideas.

---

> > ### Author Response · Authors · 2024-11-27
> > **Response to Follow-Up Questions (1/2)**
> >
> > Thank you for the follow-up queries. We have provided our responses below to address the concerns.
> >
> >
> > > The definition of "UNRELIABLE BEHAVIOR" is unclear.
> >
> > "unreliable behavior" is the term used for model's incorrect predictions that get influenced by backdoor triggers or spurious correlations. Although we believe that this understanding of the terms is quite clear from the context at multiple places, e.g., Abstract, Introduction, Related Work, we will also indicate this more explicitly. Thank you for pointing this out.
> >
> >
> > > ... why the proposed rank-1 approach can alleviate the challenges.
> >
> > Thank you for the query. There seems to be a misunderstanding. Our work does not propose a new rank-one model editing specifically to address these challenges. Instead, we leverage rank-one model editing as a mechanism to correct the model's unreliable behavior, i.e., wrong predictions due to spurious correlations or backdoors inside the model. This correction allows us to ‘sidestep’ the challenges by targeting and mitigating the root causes of the unreliable behavior. We have detailed how our method can sidestep the identified challenges in Section 4.2.
> >
> >
> > > ... no evidence supporting the claim ...
> >
> > We would like to further clarify that we do not claim that rank-one model editing inherently corrects a model's unreliable behavior. Instead, we introduce a novel methodology that leverages rank-one model editing to achieve this goal. The validity of our approach is substantiated through extensive evaluation results, which demonstrate its effectiveness in addressing unreliable behavior on corrupted or spurious inputs. Furthermore, we appreciate that the **reviewer acknowledged the strengths of our method in their original comments, describing it as "showcases extensive effectiveness."**
> >
> >
> > > ... study of UNRELIBALE BEHAIVOR data back to 2019.
> >
> > We respectfully disagree with the comment. In the very first paragraph of the paper, we clearly cite related works until 2024. If the comment is about Sec. 2, please note that the works discussed in the ‘Unreliable Model Behaviors’ paragraph are meant to cover the key references affirming/identifying the persistence of the unreliable behaviors of the models. Their relevance is in regard to providing a clear motivation to this work to address the underlying problems. Given that both spurious correlation and trojans are now well-known to the community, the key relevant works highlighting these issues are of course expected to be a few years old.
> >
> > > ...necessitates a closed form for f.
> >
> > We compute the gradients with respect to the input $x$, **relying solely on the differentiability of the function $f$**. Given that neural networks are inherently differentiable, our approach is applicable to any deep neural network model. While it is true that a closed-form expression for the deep model $f$ is not available for such models, this does not constrain the applicability of our method. We work directly with gradient information, which allows us to estimate the function disagreement effectively **without needing an explicit closed-form representation of $f$**.
> >
> >
> > > ... how this operates within neural networks ... Relying solely on the loss function in the final iteration may lead to issues.
> >
> > We appreciate the reviewer’s query. To clarify, our approach does not involve computing gradients with respect to the model weights. Instead, it focuses on the input features (e.g., input pixels). Specifically, we compute attributions as the gradients of the output logits with respect to the input features, which allows us to capture the influence of the input on the model’s predictions without relying on the loss function in the final iteration.
> >
> >
> > > ... updating x requires a trustworthy direction and appropriate step size.
> >
> > For Eq. (2), the update direction is naturally defined as $\tilde{x} \rightarrow x$, guided by the provided samples. The step size has a minimal impact on the precision of the integration estimation due to the small distance between a cleansed sample $x$ and a corrupted sample $\tilde{x}$. Furthermore, our method addresses potential concerns about step size by leveraging a verified bound on $f(x) - f(\tilde{x})$. Additionally, we draw on recent advancements, Monte-Carlo estimation, to bypass gradient computations for multiple steps [1], enhancing efficiency.
> >
> > [1] Erion, Gabriel, et al., “Improving performance of deep learning models with axiomatic attribution priors and expected gradients.” Nature Machine Intelligence (2021).

---

> > > ### Author Response · Authors · 2024-11-27
> > > **Response to Follow-Up Questions (2/2)**
> > >
> > > > ... why $\tilde{x}$ is updated using x as a given teacher, as this could result in significant problems if the update is low-confidence.
> > >
> > > In our approach, $x$ represents the cleansed sample, on which the model is expected to demonstrate robust behavior, while $\tilde{x}$ represents the corrupted sample. The update from $\tilde{x}$ to $x$ is only applied when the model's behavior can be corrected, ensuring that the update is meaningful and justifiable. This is analogous to using a training label in model optimization, where corrections are applied only when the label is considered reliable. Thus, the update does not introduce risks.
> > >
> > >
> > >
> > > > Overall,....useful ideas.
> > >
> > > Whereas we must indicate that Reviewer RTf7 particularly mentioned our paper as”‘well written” and “enjoyable to read”, we further improve the presentation along the following lines to address the reviewer’s concern.
> > > - To enhance clarity around "unreliable behavior," we explicitly define this term and discuss this concept more clearly throughout the paper, including in the abstract, introduction, methodology, and experiments sections.
> > > - To provide a clearer understanding of how our method addresses and sidesteps challenges, we reorganize Section 4.2 for better flow and explanation.
> > > - We now also included Appendix 3.5, which provides a detailed description of the attribution estimation process.

---

### Official Review · Reviewer_RTf7 · 2024-11-06

**Soundness:** 3
**Presentation:** 4
**Contribution:** 4
**Rating:** 6
**Confidence:** 4

**Summary:**

Neural network models often underperform when faced with data shifts. Due to their opaque nature, addressing this issue typically involves extensive data cleaning and retraining, resulting in significant computational and manual demands. This drives the need for more efficient model correction methods. This paper introduces a rank-one model editing approach to correct unreliable model behavior on corrupted inputs, aligning it with performance on clean data. The proposed method uses an attribution-based technique to identify the primary layer contributing to the model's misbehavior, incorporating this layer localization into a dynamic model editing process. This enables adaptive adjustments during editing. The authors performed extensive experiments which show that that their method effectively corrects issues related to neural Trojans and spurious correlations with as little as a single cleansed sample.

**Strengths:**

The paper is quite well written. The problem that is addressed is clearly defined and seems quite relevant for this venue. The proposed method has shown compelling results against all the baselines. Overall, this is an enjoyable paper to read.

**Weaknesses:**

The key limitations of this method include its reliance on identifying unreliable behaviors and the requirement that both corrupted and cleansed samples are available for effective correction. While the method has shown compelling results on almost all the benchmarks, this heavy reliance on the identification of unreliabilities makes the method less practical. Unfortunately, the authors didn't provide any clues or research directions for how to mitigate this issue. Also, it seems to me that this method is mostly applicable to models with simpler computational graphs. For instance, models that involve lots of skip connections, group norm, layer norm, etc. might be quite difficult to correct. It is also not clear to me how effective the proposed method is when dealing with stronger more  "aggressive" poisonous attacks. Mentioning how severe the attacks that are considered are and how they compare with other types of attacks might convince the reader more about the efficacy of the method.

**Questions:**

1. What are the possible research avenues to mitigate some of the limitations highlighted above?
2. How practical is your method for highly complex networks with very intricate computational graphs?
3. How severe are the poisonous attacks that are considered? Are there newer and more severe attacks that could evade your method?

---

> ### Author Response · Authors · 2024-11-18
> **Response to Reviewer RTf7**
>
> We thank the reviewer for the thoughtful feedback. We have provided our response below and hope it addresses your concerns.
>
> > … the reliance on the identification of unreliabilities …
>
> While our method relies on the identification of unreliable behaviors, this reliance is reasonably practical. In fact, as compared to existing methods, it is more practical. Specifically, our method offers the following advantages:
>
>  - **Reduced reliance on cleansed samples.** Our approach requires only a single pair of corrupted and cleansed samples to achieve effective correction. This makes it adaptable to scenarios where extensive cleansed datasets are unavailable, enabling robust model editing even in resource-constrained settings.
>  - **No need for precise trigger information.** Unlike methods that depend on exact pinpointing of backdoor triggers or spurious features, our method operates at the image level. This bypasses the need for pixel-level precision, reducing the complexity and effort associated with image cleansing while maintaining strong correction performance.
>
> Enabling model editing with only coarse detection of anomaly, our method remains practical. In Appx. 8, we discuss the generalization capabilities of our method, and we will emphasize additional insights and future research directions to enhance its applicability in broader contexts.
>
> > … for highly complex networks …
>
> Our method is extendable to complex networks. The concern regarding the applicability of our method to models with intricate computational graphs aligns with the motivation of our attribution-based layer localization approach. Attribution methods are designed to trace output changes back to specific architectures within the model, enabling us to identify the layers or modules most responsible for unreliable behavior. This identification mitigates the need for exhaustive modifications, making our approach suitable for complex architectures.
>
> > … dealing with stronger poisonous attacks …
>
> We appreciate the reviewer’s feedback and agree that clarifying the severity of the attack would strengthen our argument. Unlike newer attack methods that focus on stealth, such as those using minimal perturbations to evade detection, the blend attack embeds triggers directly into the input, maximizing the attack's impact on the model's predictions.
>
> The blend attack’s capacity to strongly bias the model’s output toward the target class underscores its severity as a threat. Given this, our method’s demonstrated robustness against such attacks provides strong evidence of its efficacy. Moreover, we believe that our approach would generalize effectively to newer or more aggressive attack methods, as they typically balance between stealth and potency in ways the blend attack already represents.
>
> To address the reviewer’s concern, we will include additional content in the revision to explain our rationale for selecting the blend attack and to compare its severity with other types of attacks.

---

> > ### Comment · Reviewer_RTf7 · 2024-11-26
> > **Post rebuttal**
> >
> > I would like to thank the authors for their rebuttal. I will maintain my score.

---

> > > ### Author Response · Authors · 2024-11-27
> > >
> > > Thank you for reviewing our rebuttal and for maintaining your score! We appreciate your thoughtful feedback.

---

### Author Response · Authors · 2024-11-20
**General Response**

We sincerely thank all the reviewers for their valuable and constructive feedback. In response to the comments, we have revised the manuscript, incorporating updates to both the main paper and the Appendix in line with our claims. The revised content is marked in blue for clarity. We also greatly appreciate the reviewers' additional suggestions and look forward to further discussions regarding our responses.

---

### Comment · Area_Chair_P3PA · 2024-11-24

Dear Reviewers,

The public discussion phase is ending soon, and active participation is highly appreciated and recommended. Thanks for your efforts and contributions.

Best regards,

Your Area Chair

---

### Author Response · Authors · 2024-11-27

We thank the reviewer for their valuable suggestions and continued engagement, which have been instrumental in improving our work. In the initial stages of revision, we used highlighted text to clearly indicate changes made in response to the reviewer’s comments. As the revision deadline approaches, to ensure the updated submission reflects a polished and cohesive document, we have removed the highlights in the latest version. The revisions themselves are retained for fully addressing the points raised in the review.

We hope this explanation provides clarity regarding the updates and ensures the revised document meets your expectations. We are also happy to engage in further discussions during the remaining discussion period.

For your convenience, we summarize the revisions below:
- Section 4.1 further clarifies the notions and symbols used.
- Section 4.2 now provides a more rigorous analysis of how our method sidesteps the identified challenges, improving flow and explanation.
- Section 5.2 now includes an analysis of the time complexity of our algorithm.
- Section 7 now includes a discussion of limitations and future work.
- The term "unreliable behavior" is now explicitly defined and discussed throughout the paper, including in the abstract, introduction, methodology, and experiments sections.
- Appendix 3.2 now provides a discussion regarding the rationale for selecting the blend attack in our evaluation.
- Appendix 3.5 now includes a detailed description of the attribution estimation process.
- Appendix 7 now further extends experiments on the Waterbirds dataset.
- Typos have been checked and corrected throughout the paper.

---

### Meta-Review · Area_Chair_P3PA · 2024-12-17

**Metareview:**

This paper focuses on mitigating the spurious correlations learned by the model during training. Motivated by this, a rank-one model editing approach is proposed to correct unreliable model behavior. An attribution-based technique is used to identify the primary layer related to the misbehavior first and then adaptive edit the model. All reviewers generally agreed that the research problem is important, and the experimental results were satisfactory and convincing. However, reviewers raised concerns about the novelty over existing methods and the lack of insightful theoretical analysis. Some feel the contribution is overclaimed and the research problem is narrow. At the end of the rebuttal, more than one reviewer was still concerned about novelty, scope, and writing. Thus this paper can not be accepted by ICLR in its current version.

**Additional Comments On Reviewer Discussion:**

At the end of the rebuttal, more than one reviewer was still concerned about novelty, scope, and writing. The remaining concerns are not minor, putting this paper below the acceptance bar.

---

### Decision · Program_Chairs · 2025-01-22

Reject